# Particle shape accounts for instrumental discrepancy in ice core dust size distributions

Marius Folden Simonsen[1], Llorenç Cremonesi[2], Giovanni Baccolo[4], Samuel Bosch[1], Barbara Delmonte[4], Tobias Erhardt[3], Helle Astrid Kjær[1], Marco Potenza[2], Anders Svensson[1], and Paul Vallelonga[1]

[1]Centre for Ice and Climate, Niels Bohr Institute, University of Copenhagen, Copenhagen, Denmark
[2]Department of Physics, University of Milan and National Institute for Nuclear Physics (INFN), Via Celoria 16, I20133 Milan, Italy
[3]Climate and Environmental Physics, Physics Institute & Oeschger Centre for Climate Change Research, University of Bern, Sidlerstrasse 5, 3012 Bern, Switzerland
[4]Department of Earth and Environmental Sciences, University Milano-Bicocca, Piazza della Scienza 1, I20126 Milan, Italy

*Correspondence to:* Marius Folden Simonsen (msimonse@fys.ku.dk)

**Abstract.** The Klotz Abakus laser sensor and the Coulter Counter are both used for measuring the size distribution of insoluble mineral dust particles in ice cores. While the Coulter Counter measures particle volume accurately, the equivalent Abakus instrument measurement deviates substantially from the Coulter Counter. We show that the difference between the Abakus and the Coulter Counter measurements is mainly caused by the irregular shape of dust particles in ice core samples. The irregular shape means that a new calibration routine based on standard spheres is necessary for obtaining fully comparable data. This new calibration routine gives an increased accuracy on Abakus measurements, which may improve future ice core record intercomparisons. We derived an analytical model for extracting the aspect ratio of dust particles from the difference between Abakus and Coulter Counter data. For verification, we measured the aspect ratio of the same samples directly using a Single Particle Extinction and Scattering Instrument. The results demonstrate that the model is accurate enough to discern between samples of aspect ratio 0.3 and 0.4 using only the comparison of Abakus and Coulter Counter data.

## 1 Introduction

Ice cores from Greenland contain a record of climate proxies over the last 120,000 years. One of those proxies is mineral dust in the size range 0.5-100 μm. The dust has several properties that provide useful information of the past: concentration, size distribution, morphology and chemical and isotopic composition. These measurements have revealed that the dust in ice cores come from central Asia during both the Holocene and the last glacial period (Biscaye et al., 1997). The observed 100-fold decrease in dust concentration from glacial to Holocene (Ruth et al., 2003; Steffensen, 1997) has constrained the aridity, windiness and insolation forcing of glacial climate models (Mahowald et al., 1999; Lambert et al., 2015).

Traditionally, the Coulter Counter technique has been used to measure concentration and size distribution. It works by measuring the electrical impedance over an orifice, through which a sample flows. For ice cores, this sample is melted ice core water, with pure NaCl added to stabilise the electrical conductivity. When a particle flows through and displaces the conductive

liquid, the impedance rises. This signal increases with the particle volume. The Coulter Counter has the disadvantage that it applies only to discrete samples and has not been combined with continuous flow analysis (CFA) systems.

CFA systems (Röthlisberger et al., 2000; Kaufmann et al., 2008) on the other hand are a common technique for analysing impurities in ice core samples, offering faster measurement speed and often higher resolution. On the Copenhagen CFA system,
35×35×550 mm sticks are cut from the ice core and melted upon a gold coated melt head (Bigler et al., 2011). The melt water from the outer 5 mm of the ice core surface is discarded, while the inner uncontaminated water is transported by a peristaltic pump to the connected instruments. One of these instruments is the Abakus laser sensor (LDS23/25bs, Klotz GmbH, Germany) for measuring insoluble particle concentration and size distribution.

The Abakus instrument measures the intensity of laser light through a flow cell filled with the sample liquid. When a
particle passes, the light is attenuated. The Abakus therefore measures the optical extinction cross section of the particle, and can measure particles in the range 1-15 μm. Since it measures optical transmissivity rather than electrical impedance, it is much less sensitive to electrical noise than the Coulter Counter. The Abakus on a CFA system can have a measurement depth resolution of 3 mm (Bigler et al., 2011), and requires almost no maintenance when the CFA system is running. The Coulter Counter on the other hand typically integrates a thicker depth interval, and requires work from the operator all the time during
measurements (Delmonte et al., 2004; Lambert et al., 2012).

The aspect ratio of ice core dust particles was measured using the novel Single Particle Extinction and Scattering (SPES) instrument (Villa et al., 2016; Potenza et al., 2016). The SPES measures both the extinction cross section, which is also measured by the Abakus, and the optical thickness of the particles (Potenza et al., 2016).

The optical thickness depends on the geometrical thickness of the particle and its refractive index. If the refractive index is
known, the aspect ratio can be derived from the combination of extinction cross section and geometrical thickness. The SPES is able to discern between oblate and prolate particles.

The extinction and scattering cross sections of irregularly shaped particles can be accurately calculated with the discrete dipole approximation (DDA) (Draine and Flatau, 1994). In the present work, we have used the Amsterdam Discrete Dipole Approximation (ADDA) code (Yurkin and Hoekstra, 2011). The ADDA simulations were used to simulate the SPES measure-
ments and thereby extract the aspect ratio from the SPES data. Furthermore the ADDA simulations were used to show that the Mie scattering effects on the optical extinction cross-section for spherical particles do not affect ice core dust due to its irregular shape (Chỳlek and Klett, 1991).

The measured samples are from the Renland Ice Cap Project (RECAP) ice core drilled during the summer of 2015 on the Renland ice cap in Eastern Greenland only 2 km away from the old Renland ice core (Hansson, 1994). This ice core covers
the last glacial cycle, and samples were taken from both the Holocene and the last glacial period (supplement A). As found for the central Greenlandic ice cores, the glacial RECAP dust comes from central Asia (Biscaye et al., 1997). The Holocene dust, similar to the old Renland core (Bory et al., 2003), is dominated by a local East Greenlandic source. The volume mode of the glacial dust is 2 μm, versus 20 μm for the Holocene dust, due to the increased transport size fractionation for the glacial dust (Ruth et al., 2003).

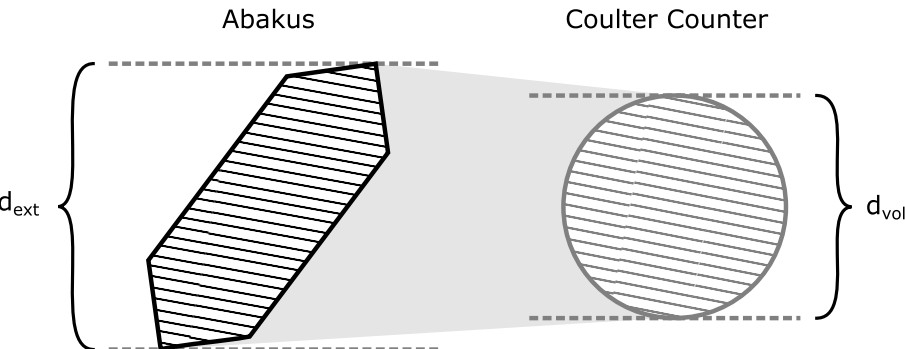

**Abakus**  **Coulter Counter**

$d_{\mathrm{ext}}$  $d_{\mathrm{vol}}$

**Figure 1.** The Abakus measures the extinction cross section of the particles, while the Coulter Counter measures their volume. Both instruments convert this to an equivalent diameter, $d_{\mathrm{ext}}$ and $d_{\mathrm{vol}}$ respectively.

Although both instruments are typically calibrated using standard spheres of known diameter, they produce substantially different size distribution results when ice core samples are measured (Ruth et al., 2003; Lambert et al., 2008, 2012; Koffman et al., 2014). It has been proposed that this difference is because ice core dust is generally non-spherical (Lambert et al., 2012; Potenza et al., 2016). We show here that the non-spherical shape of the particles does quantitatively account for the main discrepancy between the two instruments (Figure 1). One important shape effect arises from the aspect ratio, defined as the ratio between the length of its shortest and the longest side. We have found that Greenland ice core dust is dominated by oblate particles of aspect ratio $0.3 - 0.4$, which is significantly different from the aspect ratio of 1 of the polystyrene standard spheres used for calibration.

## 2   Materials and Methods

### 2.1   Abakus

For the Abakus measurements, square sticks of $35 \times 35 \times 550 \,\mathrm{mm}$ were cut from the center of the RECAP ice core, These sticks were melted on the RECAP CFA system. This CFA system is an enlargement of the one described by Bigler et al. (2011), with higher melt rate, more analytical channels and pressure decoupling after the debubbler. The Abakus was connected to the CFA system with a flow rate of $2 \,\mathrm{mL/min}$. The attached tubes and the Abakus were cleaned with MQ water (Millipore Advantage, 18.2 MOhm/cm) after each 5.5 m of ice measured. Occasionally the Abakus was clogged by a large particle and required flushing with a syringe while the system was running. The depth resolution is 5 mm due to mixing in the tubes from the melt head to the Abakus. For high dust concentrations, two particles may pass through the detector at the same time and show up as one larger particle. We found that this effect was negligible (see Supplement F for details). The Abakus was originally calibrated to give the correct diameter for polystyrene beads of 2, 5, and 10 µm.

Polystyrene standard spheres 1.5, 2, 4, 5 and 10 μm diameter from BS-Partikel GmbH, Wiesbaden, Germany (supplement B), were measured by the Abakus. They were diluted to concentrations between 15000 and 90000 particles/mL. To avoid coagulation, the standards were sonicated for 30 seconds before measurements. Each of the standards was measured for 6 minutes.

## 2.2 Coulter Counter

The Coulter Counter measured discrete samples 55 cm in length. The measured ice consisted of an outer triangular piece $3 \times 1$ cm in cross section cut to 10 cm long pieces. The samples were decontaminated by rinsing in three consecutive jars of MQ water. In each jar the outer layer of ice was melted away and removed, leaving only the cleaner inner part to be analyzed. This treatment reduced the sample size by 50%. The samples were measured in two Beckman Coulter Counters, one with a 100 μm aperture and one with a $30\,\mu$m aperture. The samples were shaken prior to measurement in the $100\,\mu$m Coulter Counter. Afterwards the samples were measured by the 30 μm Coulter Counter. The 30 μm aperture data were used for particles smaller than 4 μm and the 100 μm aperture data were used for particles larger than 4 μm. Selected samples representative of Holocene and glacial climates, were measured by the Coulter Counter for this study. For the Holocene, selected samples from 356 to 4008 years b2k (before AD 2000) were used, while for the glacial, the whole period from 17760 to 33885 years b2k was measured (supplement A). The same samples were used for the Abakus.

## 2.3 SPES

After the Coulter Counter measurements, the samples were measured by SPES (section 3.3). The sample flows through a 200 μm flow cell, and is illuminated by a laser. The light is measured by two detectors, which in combination gives the extinction cross section and the optical thickness. There is no focusing of the particle stream in the cell, so only a small fraction of the particles passing through the cell is measured by the laser. Therefore the sample is circulated through the cell several hundred to thousand times. This gives an accurate measure of the shape distribution of the particles, but it does not allow concentration measurements.

The narrow cell ensures high shear, forcing the particles to attain a preferential direction along one direction. Oblate particles in a shear flow orient themselves with the flat side along the flow lines, and are randomly oriented in the shear direction. Prolates lie in a plane of constant velocity, and are free to rotate within it (Jeffery, 1922).

## 3 Results

### 3.1 ADDA simulations

Using the Amsterdam Discrete Dipole Approximation software (ADDA) (Yurkin and Hoekstra, 2011), we have calculated the extinction diameter for different particles. The extinction diameter is based on the optical extinction cross section. The optical extinction cross section is defined for a plane light wave interacting with a particle as the difference between the incoming and transmitted light intensity divided by the incoming light intensity and multiplied by the area of the plane incoming wave.

For spherical particles much larger than the wavelength of the light, the relation between diameter and optical extinction cross section is $\sigma_{\text{ext}} = \frac{\pi}{2} d^2$. For smaller particles of size comparable to the light wavelength, the relation differs due to optical effects (Figure 2) (van de Hulst, 1957). However, we define the extinction diameter as $d_{\text{ext}} = \sqrt{\frac{2}{\pi} \sigma_{\text{ext}}}$ for all particles. We associate each particle with its volumetric diameter: A sphere with volume $V$ has the diameter $d_{\text{vol}} = \sqrt[3]{\frac{6V}{\pi}}$. For any particle of known volume, the volumetric diameter is given by this relation.

Specifically, we have used ellipsoids and oblate prisms with an aspect ratio of 0.3 in the volumetric diameter range 1 to 8 μm (Figure 3). While spheres have a unique extinction diameter for each volumetric diameter, the extinction diameter of other particles depends on their orientation. For each volumetric diameter, there will therefore be several possible Abakus measurements of the extinction diameter, described by a probability density function. This is described in more detail in Section 3.4, where it is called $\frac{dP(d_{\text{ext}}|d_{\text{vol}})}{d\ln d_{\text{ext}}}$. The broadness of the probability density function for non-spherical particles results in a smoothing of the relationship between the extinction and volumetric diameters (Figure 3). Furthermore, the extinction diameter oscillation maxima are not located at the same $d_{\text{vol}}$ for different particle shapes. Since ice core dust has a variable shape (supplement C), it is sampled from a sum of the ellipsoid, prismatic and many more distributions. In this sum of distributions the oscillation pattern averages out.

The average extinction diameter is larger than the volumetric diameter since the measured dust particles are elongated (discussed further in section 3.4). For particles larger than 1.7 μm, the average extinction diameter is approximately proportional to the volumetric diameter. For much smaller particles, the extinction diameter is independent of the particle shape. After calibration, the Abakus cannot measure particles with an extinction diameter smaller than 1.8 μm, so the measured Abakus data is within the range of proportionality: $d_{\text{ext}} \propto d_{\text{vol}}$ (Section 3.2).

## 3.2 Extinction calibration using particle size standards

We have measured the diameters of five different particle size standards (supplement B, section 2.1) with the Abakus (Figure 2). The 2, 5, and 10 μm standards were measured twice with a one year delay, without significantly different results. This shows that the Abakus calibration is stable over the time scale of a typical measurement campaign.

Since $d_{\text{ext}}$ is proportional to $d_{\text{vol}}$ for ice core dust (section 3.1), and the proportionality constant depends on particle shape and is unknown, we would like to calibrate the Abakus such that it gives the extinction diameter $d_{\text{ext}}$ and not the true diameter for the standard spheres (using the term true for the certified diameter given by the manufacturer). Therefore we define a calibration function such that the measured diameter divided by the calibration function is the extinction diameter. To avoid artifacts in the calibrated distribution, the calibration function has to be a continuous function of the measured diameter with a continuous first derivative. Since we want the relative error on each standard to have the same weight, the function is fitted to the logarithm of the ratio between the measured and extinction diameter. The function $y = a(x - x_0)^2$, where $y = \frac{\ln(d_{\text{meas}})}{\ln(d_{\text{ext}})}$ and $x = \ln(d_{\text{meas}})$, is fitted to the data as calibration function (Figure 4). The fit parameters found are $(a, x_0) = (-0.086, 2.60)$. This particular calibration function is chosen because it is simple, fits the data well and satisfies the conditions described above. $x_0 = 2.60$ corresponds to $d_{\text{meas}} = 13.5$ μm. As the calibration function is equal to 1 for $d_{\text{meas}} = 13.5$ μm and standards larger than 13.5 μm were not tested, no calibration is applied for diameters larger than 13.5 μm.

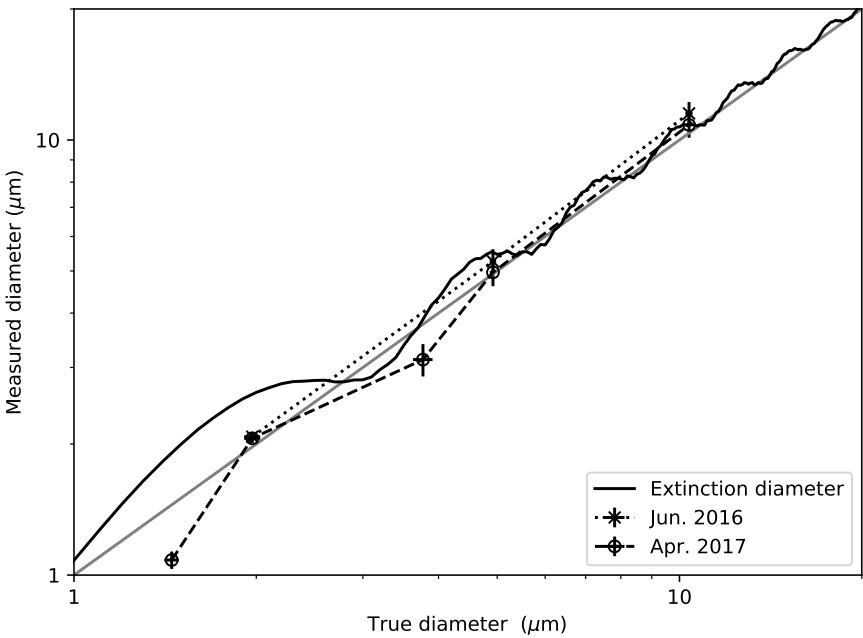

**Figure 2.** The diameter of standard polystyrene spheres measured by the Abakus as a function of the true diameter certified by the manufacturer (circles and dashed line), together with three of the same standards measured a year earlier (crosses and dotted line) and the optical extinction diameter of a sphere as a function of diameter (solid black curve). The grey line is where the measured and the true diameter are equal, to which the extinction diameter converges for large true diameters.

The particle size standard calibration has been applied to ice core dust data (orange curves in Figure 5a,b). Since the calibration function is less than 1 for diameters smaller than 13.5 μm, the effect of applying the calibration function is that the small diameter data is shifted towards larger diameters. This is most pronounced for the smallest diameters, since the calibration values are smallest for small diameters. The positive slope of the calibration function means that the calibrated bin positions are squeezed more tightly together than the uncalibrated bins. Since the probability density function is defined as the number of counts in a bin divided by its size, the calibrated probability density function has higher values than the uncalibrated one. This effect decreases with diameter, as the slope decreases. For further details, see Supplement H.

### 3.3 SPES

We have measured optical thickness ($\rho$) and extinction cross section ($\sigma_{\mathrm{ext}}$) with SPES for both Holocene and glacial samples (Figure 6), using the method of Villa et al. (2016), and compared the results to ADDA simulations. The distribution of particles in $\rho, \sigma_{\mathrm{ext}}$ space can be simulated using ADDA for a given particle shape and size distribution. We have done this for oblate right square prisms of aspect ratios 0.2, 0.25, 0.3, 0.4 and 0.5. A total of 20,000 particles with a volumetric diameter ranging from

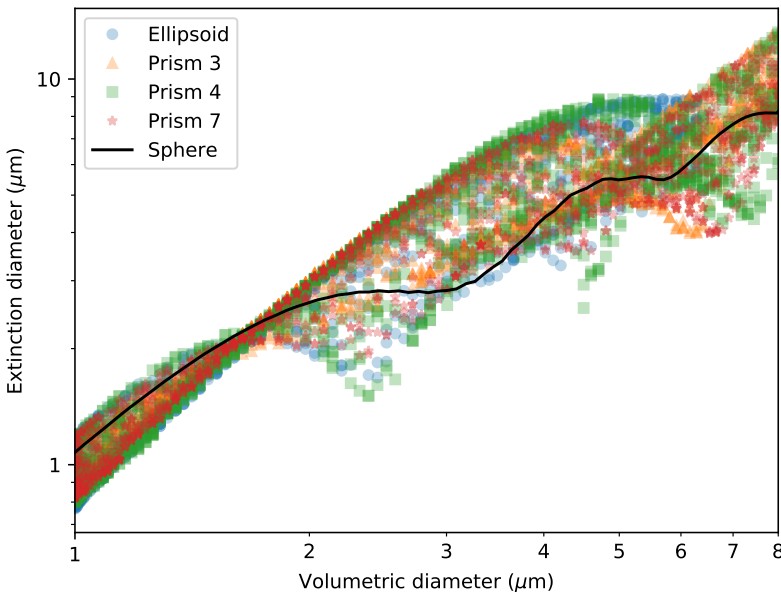

**Figure 3.** ADDA simulations of particles with a refractive index of 1.586 and, except for the spheres, an aspect ratio of 0.3. The prisms are oblate and their cross sections are equilateral polygons with 3, 4 and 7 sides.

100 nm to 2 µm and different refractive indexes and orientations were simulated. This size range covers the SPES measurement range.

The refractive index, $n$, of atmospheric dust is on average between 1.52 and 1.55 (Shettle and Fenn, 1979; Sokolik et al., 1993; Grams et al., 1974). At Dome C, Antarctica, the refractive index of Holocene and glacial ice core dust is 1.53 and 1.56 (Royer et al., 1983). We have run the simulations using the refractive indices 1.52 and 1.55 and found only a small effect of the refractive index on the modelled aspect ratio.

By comparing to SPES measurements of oblate and prolate particles in Villa et al. (2016) and Potenza et al. (2016), it was found that the samples are dominated by oblates. Prolates have a much narrower distribution of optical thickness than oblates, since their orientation is fixed by the flow. The absence of a superimposed prolate distribution indicates that no more than 15% prolates are compatible with the measured SPES results. The following analysis therefore only focuses on oblates. For a similar analysis of prolates, see supplement G.

To compare the simulations and the data, the average of the logarithm of the simulated optical thickness as a function of extinction cross section was calculated. This average was then used as a least squares fit to the measured data, where the aspect ratio is the variable parameter. For values of $\sigma_{\mathrm{ext}}$ approaching the lower and upper bound of the extinction cross section range, the SPES is not sensitive in the full optical thickness range. To avoid bias, only experimental data between the 0.25 and 0.75 quantile of the extinction cross section was used in the fit.

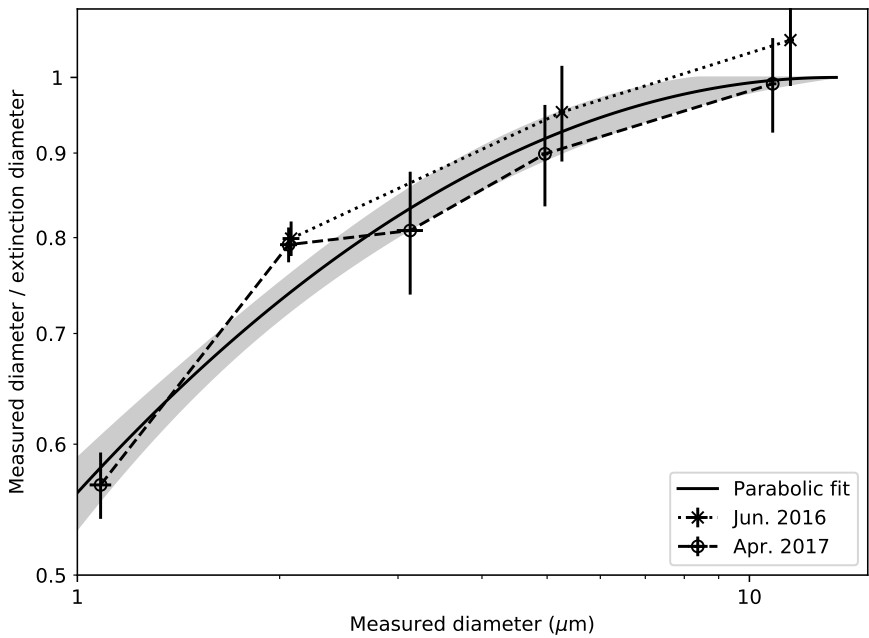

**Figure 4.** The ratio between the diameter of spheres measured by the Abakus in April 2017 and their extinction diameter (Figure 2) (circles and dashed line), together with three of the same standards measured a year earlier (crosses and dotted line) and a fit to the logarithm of the April 2017 data (solid line). The fitted parameters are $(a, x_0) = (-0.086, 2.60)$ of the function $a(x - x_0)^2$ where $x$ is the logarithm of the diameter in μm. The uncertainty on the fit (shading) is based on the uncertainty on the data points.

For $n = 1.52$ the Holocene aspect ratio is $0.42 \pm 0.03$ and the glacial is $0.38 \pm 0.02$, while for $n = 1.55$ the Holocene is $0.36 \pm 0.01$ and the glacial is $0.29 \pm 0.01$. This is independent of the volumetric diameter size distribution used in the simulation. The average in the refractive index range of ice core dust is therefore $0.39 \pm 0.03$ for the Holocene and $0.33 \pm 0.04$ for the glacial.

## 3.4 Aspect ratio effect

Spheres have the lowest geometrical cross section to volume of all particles, when averaged over all rotation angles of the particles (Brazitikos et al., 2014). We define the geometric cross section diameter of a particle as $d_{\text{geom}} = \sqrt{\frac{2}{\pi} \sigma_{\text{geom}}}$, where $\sigma_{\text{geom}}$ is the geometric cross section of the particle. However, since ice core dust is non-spherical, (section 3.3), $d_{\text{geom}} > d_{\text{vol}}$ and $d_{\text{ext}} = d_{\text{geom}}$ (section 3.2). We can therefore calculate the distribution of $d_{\text{ext}}$ based on the aspect ratio $c$ and the distribution of $d_{\text{vol}}$. The distribution of $d_{\text{vol}}$, $\frac{dP(d_{\text{vol}})}{d \ln d_{\text{vol}}}$, can be determined by the Coulter Counter.

Since oblates dominate the measured samples, we will focus our model on those, instead of also considering prolates (section 3.3). As previously mentioned (section 2.3), oblate particles in a shear flow orient themselves with the flat side along the flow lines, while they are randomly oriented in the shear direction (Jeffery, 1922). Since they are free to rotate along an axis

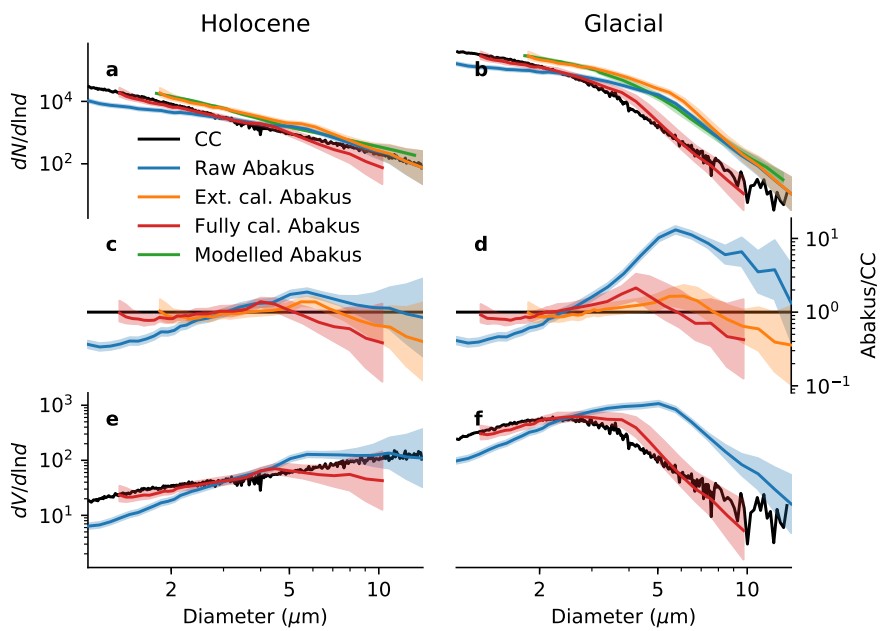

**Figure 5.** Size distributions of the dust particles in the RECAP ice core for both Holocene and last glacial period measured by Abakus and Coulter Counter. **a,b**: Probability density functions of the number of particles as function of volumetric diameter, measured diameter or extinction diameter for Coulter Counter (black, volumetric diameter), raw Abakus (blue, measured diameter), extinction calibrated Abakus (Section 3.2) (orange, extinction diameter), modelled Abakus data based on Coulter Counter data and the aspect ratio measured by SPES (Section 3.4) (green, extinction diameter) and Abakus data fully calibrated to Coulter Counter data using both the extinction calibration and aspect ratio (Section 3.5) (red, volumetric diameter). The uncertainty (shaded area) on the raw Abakus data is measurement uncertainty (Supplement E), while the remaining uncertainties are linearly propagated from the Abakus and aspect ratio uncertainties. **c,d**: Raw Abakus data divided by Coulter Counter (blue, from blue and black in a, b), extinction calibrated Abakus data divided by modelled Abakus data (orange, from orange and green in a, b) and fully calibrated Abakus data divided by Coulter Counter (red, from red and black in a,b). **e,f**: Probability density functions of particle volume derived from the number density functions of figure a,b for Coulter Counter (black), raw Abakus (blue) and fully calibrated Abakus data (red).

orthogonal to the light beam direction, we can model them as rectangles embedded in 2 dimensions for which all orientation angles are equally likely (supplement D). The light and detector also lie in the plane. In this 2 D model, $d_{\mathrm{vol}}$ is the square root of the area of the rectangle, and $d_{\mathrm{ext}}$ is the cross section of the rectangle.

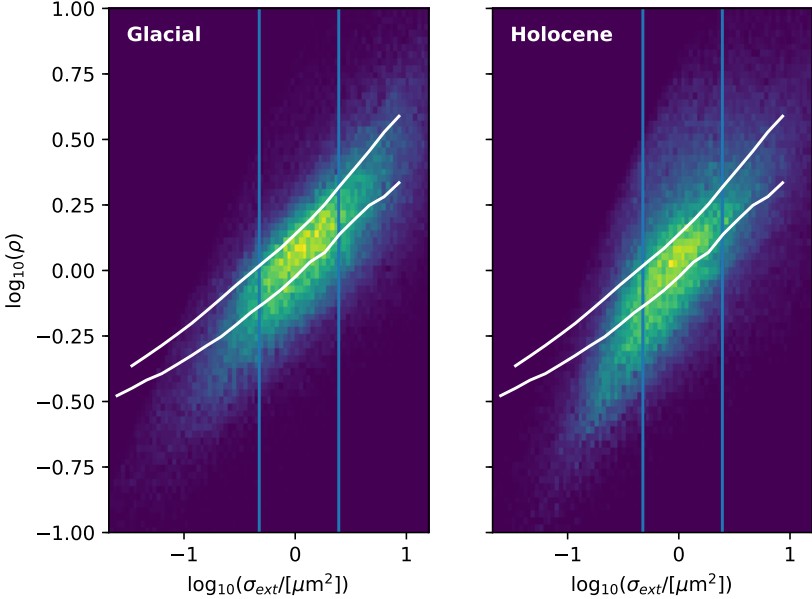

**Figure 6.** Glacial and Holocene samples measured by SPES. The brighter colors have a higher number of measured particles in a bin. The blue lines mark the 0.25 and 0.75 quantiles of $\sigma_{\text{ext}}$. The white curves are the mean optical thickness as a function of $\sigma_{\text{ext}}$ for ADDA simulations with a refractive index of 1.55. The upper is for an aspect ratio of 0.5, the lower for 0.2.

Denoting the length of the short side $b$, the length the long side $a$, and the aspect ratio $c = b/a$, the probability of $d_{\text{ext}}$ given a certain $d_{\text{vol}}$ is

$$\frac{dP(d_{\text{ext}}|d_{\text{vol}})}{d\ln d_{\text{ext}}} = \begin{cases} 0 & \text{for } d_{\text{ext}} < b \\ z & \text{for } b < d_{\text{ext}} < a \\ 2z & \text{for } a < d_{\text{ext}} < \sqrt{a^2 + b^2} \end{cases}, \tag{1}$$

for

$$z = \frac{2}{\pi} \frac{1}{\sqrt{\frac{a^2+b^2}{d_{\text{ext}}^2} - 1}}. \tag{2}$$

The distribution of $d_{\text{ext}}$ for all $d_{\text{vol}}$ can then be found as

$$\frac{dP(d_{\text{ext}})}{d\ln d_{\text{ext}}} = \int \frac{dP(d_{\text{ext}}|d_{\text{vol}})}{d\ln d_{\text{ext}}} \frac{dP(d_{\text{vol}})}{d\ln d_{\text{vol}}} d\ln d_{\text{vol}}. \tag{3}$$

In section 3.2 we calibrated the Abakus such that $d_{\text{meas}} = d_{\text{ext}}$. Therefore $\frac{dP(d_{\text{ext}})}{d\ln d_{\text{ext}}}$ calculated here simulates Abakus measurements.

This can be used to find the aspect ratio of the particles just from the Coulter Counter-Abakus discrepancy. To do this, the sum of the square of the logarithm of the ratio between $\frac{dP(d_{ext})}{d\ln d_{ext}}$ calculated from the Coulter Counter data and the Abakus data was minimized with respect to the aspect ratio. This gives the aspect ratio where $\frac{dP(d_{ext})}{d\ln d_{ext}}$ is most consistent with the Abakus data, which is the most likely aspect ratio given the data. For the Holocene data this gave $c = 0.41 \pm 0.09$, while for the glacial $c = 0.31 \pm 0.04$. The errors are propagated from the total errors on the calibrated Abakus data. There is however a large correlation between the errors on the Holocene and glacial data, so the error on the difference is only around 0.02, confirming a significant difference in aspect ratio between glacial and Holocene dust particles.

## 3.5 Calibration of Abakus

Equation 3 gives the extinction diameter size distribution (Abakus) given a measured volumetric size distribution (Coulter Counter) and a known aspect ratio. However, often a measurement of the volumetric size distribution is desired, while only Abakus measurements are available. This requires the inversion of Equation 3, which cannot be done analytically. However, multiplying the extinction calibrated Abakus bins by the cubic root of the aspect ratio is a good approximation. This gives the fully calibrated Abakus data of Figure 5. For the glacial volumetric distributions, the modes of the Coulter Counter, the uncalibrated Abakus and the fully calibrated Abakus data are respectively 2.3, 5.0 and 2.7 μm. Further details on the calibration are found in Supplement H.

## 4 Discussion

Ruth et al. (2008, Figure 1a) demonstrated that the data produced by the Coulter Counter and Abakus are proportional over four orders of magnitude, even if the absolute concentration results do not agree. This is partly because the instruments have different detection limits (see Supplement I), and partly because they measure two different properties of the particles: volumetric and extinction cross section. When the Abakus is calibrated using the true diameter of polystyrene spheres, it gives up to 10 times as many counts as the Coulter Counter for some particle sizes, when ice core samples are measured (Figure 5). It is not possible in general to calibrate the Abakus such that it yields the same distribution as the Coulter Counter for all ice core samples. However, by combining Coulter Counter and Abakus data, additional information is gained about the aspect ratio that was not available from the two instruments individually.

It has previously been suggested that it is impossible to calibrate the Abakus using polystyrene beads due to the strong Mie oscillations (Ruth, 2002, p. 20). It was argued that since for spheres the measured extinction cross section is a non-monotonous function of the true diameter, it cannot be inverted. However, this is based on the criterion that $d_{meas}$ of the Abakus should be identical to $d_{true}$ when measuring spheres. Since spheres have strong Mie oscillations but ice core dust does not, this criterion is invalid. Therefore the Abakus should be calibrated such that $d_{meas}$ is equal to $d_{ext}$ instead of $d_{true}$ (section 3.2).

With the SPES instrument we found that the average aspect ratio of our ice core dust samples is $0.39 \pm 0.03$ for the Holocene and $0.33 \pm 0.04$ for the glacial, so the particles are significantly elongated (section 3.3). Using our simple model for the relation between $d_{meas}$ and $d_{vol}$, we have calculated the aspect ratio independently from the Abakus and Coulter Counter data. This

gave an aspect ratio of $0.41 \pm 0.09$ for the Holocene and $0.31 \pm 0.04$ for the glacial in accordance with SPES. The SPES aspect ratio was calculated for particles less than 2 μm volumetric diameter, and the Abakus for particles between 1.2 and 9 μm, so the measurements are not directly comparable. However, as we only investigate the leading order effect of the aspect ratio, and atmospheric studies find no size dependence of the aspect ratio (Knippertz and Stuut, 2014, page 28), it is assumed that any possible size dependence of the aspect ratio is not large enough to significantly change the results. In addition to giving the same aspect ratio as SPES, the model also gives a consistent size distribution within the Abakus uncertainties (Figure 5).

In the Dome C ice core from the East Antarctic plateau the aspect ratios of oblate and prolate particles have been determined to be $0.2 \pm 0.1$ and $3.5 \pm 1.3$, respectively (Potenza et al., 2016). The ranges refer to the variability in the aspect ratio among different particles and not the uncertainty of the mean. In the RECAP ice core studied here, we found a similar but slightly less extreme aspect ratio both in the Holocene and in the glacial ice. For the RECAP core, we speculate that the Holocene dust originates from local Eastern Greenlandic sources, while the glacial dust is from central Asia (Bory et al., 2003). Measurements of dust particles in dust storms generally show an aspect ratio above 0.5. However, during transport, the dust fractionates towards more extreme aspect ratios (Knippertz and Stuut, 2014, p. 28-30). Since large ice sheets are located far from typical dust sources, the dust extracted from ice cores would be subject to more fractionation. Therefore we do expect more extreme aspect ratios to be found in ice core dust than in the atmospheric dust storm measurements. The greater aspect ratio of the local Greenlandic Holocene dust compared to the Asian glacial dust agrees well with the observed aspect ratio fractionation observed in the atmosphere.

## 5 Conclusions

The Abakus laser sensor and the Coulter Coulter give different size distributions when measuring the same ice core dust sample. This is because ice core dust particles are not spherical, so the particle volume measurements of the Coulter Counter are different from the cross section measurements of the Abakus. When spherical particles are measured by the Abakus, the measured extinction diameter oscillates strongly as a function of particle size due to Mie scattering oscillations. The extinction diameter of ice core dust does not show this oscillation pattern, but is proportional to the volumetric diameter. When the Abakus is calibrated using spherical particles, it should therefore be calibrated to the extinction diameter and not to the true diameter.

We derived a model for extracting the aspect ratio of the dust particles from the differences between Abakus and Coulter Counter measurements of the same ice core dust samples. This process suggests an aspect ratio of $0.41 \pm 0.01$ for Holocene and $0.32 \pm 0.01$ for glacial dust samples from the RECAP ice core, consistent with direct aspect ratio measurements from a Single Particle Extinction and Scattering instrument. This shows that not only is the discrepancy between the two instruments explained by the non-spherical shape of the particles, it can also be used to obtain the aspect ratio. As the Holocene dust has Greenlandic origin while the glacial dust is Asian, the aspect ratio could potentially aid in provenance determination and in understanding atmospheric transport processes. Moreover, by determining the aspect ratio, a better size distribution can be obtained from the Abakus data.

## 6   Data availability

The data for all plots will be made available on www.iceandclimate.dk/data upon acceptance.

## Appendix A:  Bags measured by the Coulter Counter

| Holocene | | | | |
|---|---|---|---|---|
| Bag | Top depth (m) | Bottom depth (m) | Top age (years b2k) | Bottom age (years b2k) |
| 300 | 164.45 | 165.00 | 356 | 358 |
| 304 | 166.65 | 167.20 | 363 | 365 |
| 305 | 167.20 | 167.75 | 365 | 367 |
| 321 | 176.00 | 176.55 | 395 | 397 |
| 335 | 183.70 | 184.25 | 423 | 425 |
| 714 | 392.15 | 392.70 | 1980 | 1989 |
| 716 | 393.25 | 393.80 | 1998 | 2006 |
| 746 | 409.75 | 410.30 | 2282 | 2293 |
| 747 | 410.30 | 410.85 | 2293 | 2303 |
| 755 | 414.70 | 415.25 | 2379 | 2390 |
| 759 | 416.90 | 417.45 | 2423 | 2434 |
| 760 | 417.45 | 418.00 | 2434 | 2446 |
| 762 | 418.55 | 419.10 | 2457 | 2469 |
| 788 | 432.85 | 433.40 | 2785 | 2799 |
| 795 | 436.70 | 437.25 | 2884 | 2899 |
| 797 | 437.80 | 438.35 | 2914 | 2928 |
| 799 | 438.90 | 439.45 | 2943 | 2958 |
| 800 | 439.45 | 440.00 | 2958 | 2974 |
| 836 | 459.25 | 459.80 | 3588 | 3608 |
| 837 | 459.80 | 460.35 | 3608 | 3629 |
| 838 | 460.35 | 460.90 | 3629 | 3650 |
| 840 | 461.45 | 462.00 | 3671 | 3692 |
| 842 | 462.55 | 463.10 | 3713 | 3734 |
| 847 | 465.30 | 465.85 | 3822 | 3845 |
| 848 | 465.85 | 466.40 | 3845 | 3868 |
| 850 | 466.95 | 467.50 | 3890 | 3914 |
| 851 | 467.50 | 468.05 | 3914 | 3937 |
| 852 | 468.05 | 468.60 | 3937 | 3961 |
| 854 | 469.15 | 469.70 | 3984 | 4008 |

| Glacial | | | | |
|---|---|---|---|---|
| Bag | Top depth (m) | Bottom depth (m) | Top age (years b2k) | Bottom age (years b2k) |
| 972 | 534.05 | 534.60 | 17760 | 21170 |
| 973 | 534.60 | 535.15 | 21170 | 24745 |
| 974 | 535.15 | 535.70 | 24745 | 28783 |
| 975 | 535.70 | 536.25 | 28783 | 33885 |

## Appendix B:  Polystyrene sphere standards

The polystyrene sphere standards were produced by BS-Partikel GmbH, Wiesbaden, Germany. $d_{\mathrm{true}}$ is the certified diameter given by the manufacturer, $d_{\mathrm{meas}}$ is the diameter measured by the Abakus, and $\sigma_{\mathrm{true}}$ and $\sigma_{\mathrm{meas}}$ are their uncertainties.

| Nominal diameter (µm) | $d_{\mathrm{true}}$ (µm) | $d_{\mathrm{meas}}$ (µm) | $\sigma_{\mathrm{true}}$ (µm) | $\sigma_{\mathrm{meas}}$ (µm) | Lot No. | Catalog No. |
|---|---|---|---|---|---|---|
| 1.5 | 1.45 | 1.08 | 0.04 | 0.05 | LS0248.161 | LS0150-05 |
| 2 | 1.97 | 2.06 | 0.06 | 0.05 | LS239.111 | LS0200-05 |
| 4 | 3.77 | 3.12 | 0.14 | 0.27 | LS237.161 | LS0400-05 |
| 5 | 4.92 | 4.96 | 0.06 | 0.35 | LS122.111 | LS0500-05 |
| 10 | 10.37 | 10.83 | 0.14 | 0.71 | LS108.509 | LS1000-05 |

## Appendix C:  Microscope photographs

Photographs of both glacial and Holocene dust were taken through a microscope (Figures 7 and 8). The highly non-spherical shape is clearly seen. As the particles orient themselves with as low center of mass as possible on the substrate, they will typically lie on their flattest side. The aspect ratio is therefore not directly visible.

## Appendix D:  Extinction diameter calculation in the 2D model

The particles in the Abakus are modelled as 2D rectangles, for which all rotation angles are equally likely (Figure 9). The cross section of the particle is equal to $d_{\mathrm{ext}}$ in the 2D model.

## D1   Rod

For calculating the cross section of a rectangle, the cross section of a rod is needed. We define a rod with length $l$ and zero width. For an angle of rotation $\phi$, the cross section is $d_{\mathrm{ext}} = l\sin\phi$. From symmetry, the $\phi$ values are confined to $\phi \in [0, \frac{\pi}{2}]$ Assuming a uniform probability distribution for the angle, the probability of measuring the particle in the interval $[\phi, \phi + d\phi]$ is $dP(\phi; d\phi) = \frac{2}{\pi}d\phi$, which is normally written $\frac{dP(\phi)}{d\phi} = \frac{2}{\pi}$.

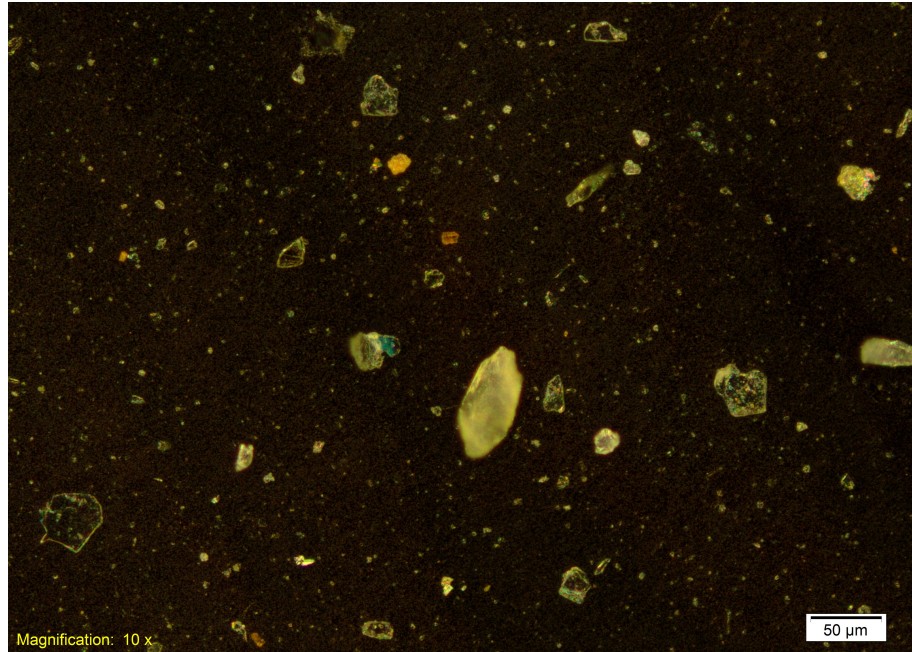

**Figure 7.** Microscope photograph of RECAP Holocene dust from bag 837.

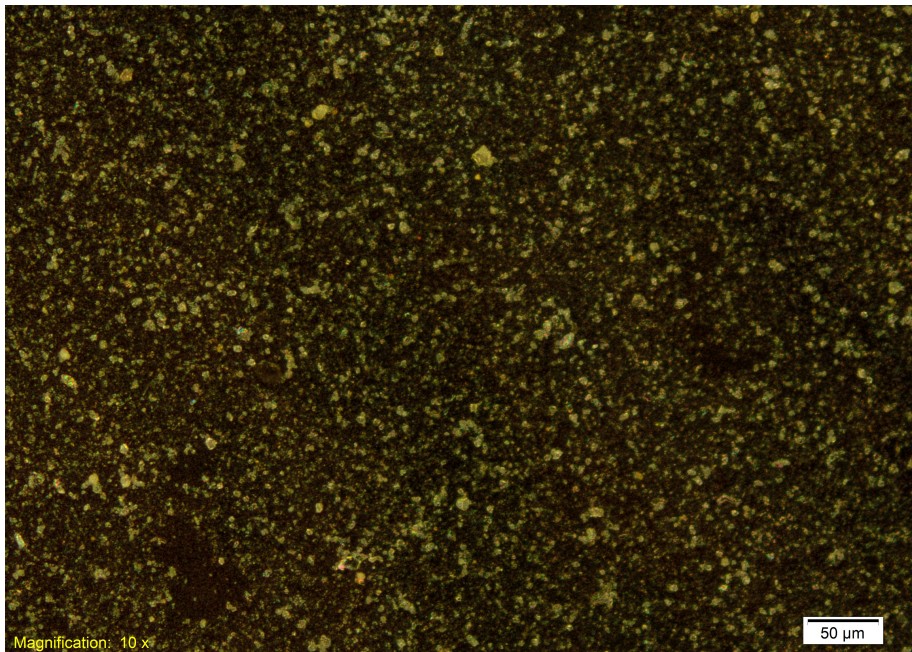

**Figure 8.** Microscope photograph of RECAP glacial dust from bag 975.

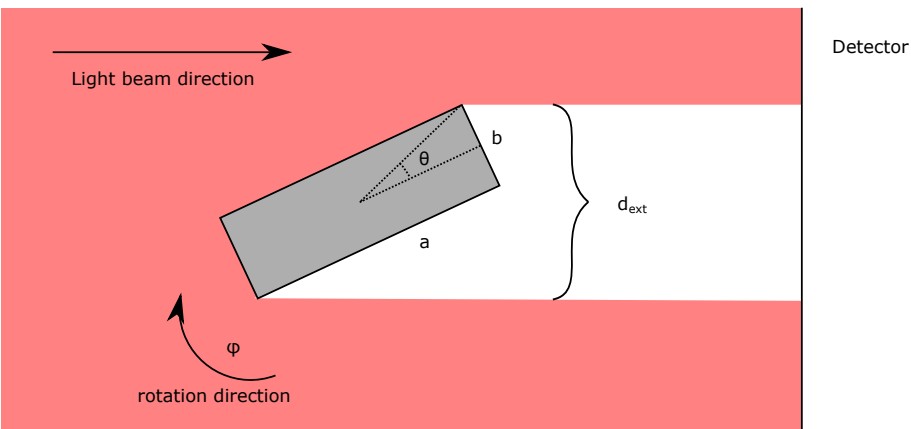

**Figure 9.** 2D model of the particle orientation in the Abakus.

10    For converting this to $d_\text{ext}$ space, $d\phi$ needs to be expressed in terms of $dd_\text{ext}$:

$$d_\text{ext} = l \sin \phi \tag{D1}$$

$$\Rightarrow dd_\text{ext} = l \cos \phi \, d\phi \tag{D2}$$

$$\Rightarrow d\phi = \frac{1}{\sqrt{l^2 - d_\text{ext}^2}} dd_\text{ext} \tag{D3}$$

Therefore

$$\frac{dP(d_\text{ext})}{dd_\text{ext}} = \frac{2}{\pi} \frac{1}{\sqrt{l^2 - d_\text{ext}^2}} \tag{D4}$$

$$\Rightarrow \frac{dP(d_\text{ext})}{d \ln d_\text{ext}} = z, \tag{D5}$$

where

$$z = \frac{2}{\pi} \frac{1}{\sqrt{\left(\frac{l}{d_\text{ext}}\right)^2 - 1}}. \tag{D6}$$

It is seen that it diverges for $d_\text{ext} \to l$, because $d_\text{ext}$ is almost constant as a function of angle when the rod is close to being perpendicular to the light source.

## D2   Rectangle

5    Consider a rectangle with side lengths $a$ and $b$. As for the rod, by symmetry, it is only necessary to consider the angles $\phi \in [0, \frac{\pi}{2}]$. In this case the cross section is given by the cross section of the diagonal. This is just the cross section of a rod rotated by

$$\phi_+ = \phi + \theta \tag{D7}$$

$$\Rightarrow \phi_+ \in [\theta, \frac{\pi}{2} + \theta], \tag{D8}$$

where $\theta$ is the angle between the side $a$ and the diagonal. If $\phi$ is uniformly distributed in $[0, \frac{\pi}{2}]$, so is $\phi_+$ in $[\theta, \frac{\pi}{2} + \theta]$. The derivation for the rod is only valid when $d_{ext}$ is a monotonous function of $\phi$. This means that it can only be directly applied for $\phi_+ < \frac{\pi}{2}$. However, due to symmetry, the cross section is the same for $\phi_+ = \frac{\pi}{2} + \Delta\phi$ and $\phi_+ = \frac{\pi}{2} - \Delta\phi$, for any $\Delta\phi$. Therefore, the probability of measuring a cross section corresponding to $\phi \in [\frac{\pi}{2} - \theta, \frac{\pi}{2}]$ is twice as high as the probability of measuring the cross section of a rod in this interval. The cross section of $\frac{\pi}{2} - \theta$ is $a$. This means that the probability distribution is

$$\frac{dP(d_{ext})}{d\ln d_{ext}} = \begin{cases} 0 & \text{for } d_{ext} < b \\ z & \text{for } b < d_{ext} < a \\ 2z & \text{for } a < d_{ext} < \sqrt{a^2 + b^2} \end{cases}, \tag{D9}$$

for

$$z = \frac{2}{\pi} \frac{1}{\sqrt{\frac{a^2+b^2}{d_{ext}^2} - 1}}. \tag{D10}$$

## Appendix E: Replicate ice core measurements

The upper 93 m of the RECAP ice core has been measured twice (Figure 10). This means that two parallel sticks have been cut from the core, each of which has been measured on the Copenhagen CFA system. The CFA system was modified slightly between the two measurement campaigns, including changes in the pump and tubing setup, as well as general wear and roughening of the tubing walls. Therefore, the difference between the two measurements most likely reflects the error introduced by the CFA system. For the smaller particles, the replicate values are 12 % larger than the main. In comparison, for the large particles the replicate measurement has up to 3 times lower values than the main measurement. This is probably because the transport of the large particles from the melt head to the instrument depends more on the specific system setup than the small particle transport.

The difference between the two measurements is used as uncertainty for the Abakus measurements. However, a mininum uncertainty of 12 % is used, since the crossing of the two curves gives an artificially low differerence.

## Appendix F: Coincidence

There is no indication of coincidence in the Abakus. If two small particles coincide in the Abakus detector, their combined shadow will make them look like a larger particle. This effect might skew the Abakus distribution towards showing more large particles. It has previously been shown that the coincidence effect is negligible for concentrations below 240,000 particles/mL (Saey, 1998). The highest dust concentrations measured in the RECAP core were around 220,000 particles/mL. We will therefore test whether there is a coincidence bias in the following way. If the concentration of small particles is $C$, the concentration of coincidences is proportional to $C^2$. In an ice core dust sample, we would expect the ratio between small and large particles to be more or less independent of the concentration. We can therefore check for coincidence by comparing the ratio

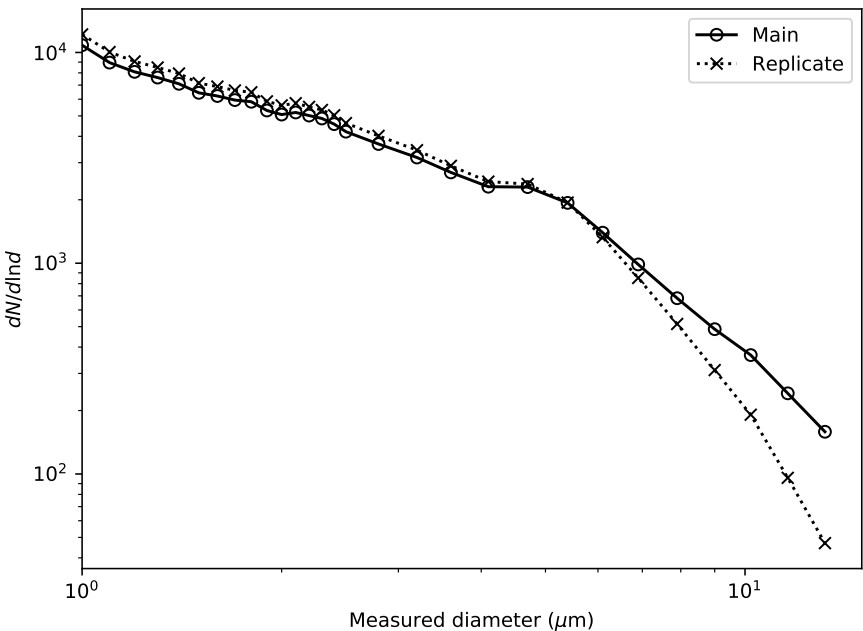

**Figure 10.** The upper 93 m of the RECAP ice core measured twice with the Abakus.

20  between small and large particle concentrations for different total dust concentration. In the glacial, for dust concentrations over 200 ppb, there is no correlation between the relative large particle concentration and the total concentration (figure 11). For concentrations smaller than 200, the relative large particle concentration is slightly larger. This could be a climatic signal. We find no significant trend with respect to concentration, suggesting no significant coincidence effect for our measured ice core samples.

25  **Appendix G: Prolate particles**

The RECAP samples consist primarily of oblate particles, so the analysis of sections 3.3 and 3.4 have not included prolate particles. The extension to prolate particles is however straight forward, and we show it here for possible future samples dominated by prolates. The aspect ratio of the RECAP samples will now be derived by comparing real Abakus data and modelled Abakus data derived from Coulter Counter data under the assumption that the particles are prolate. Prolate particles are free to rotate in a plane of constant velocity, but cannot rotate out of the plane (Jeffery, 1922). To describe their orientation, define a cartesian coordinate system with $z$ in the flow direction, $x$ in the laser light direction, and origin at the center of the

5  rectangle (Figure 12). If a prolate particle is located at $y = 0$ and $x \neq 0$, its long side will always face the light beam, since its rotation is in the plane orthogonal to the light beam. In the 2D rectangle model (Figure 9), this corresponds to the side $a$ always

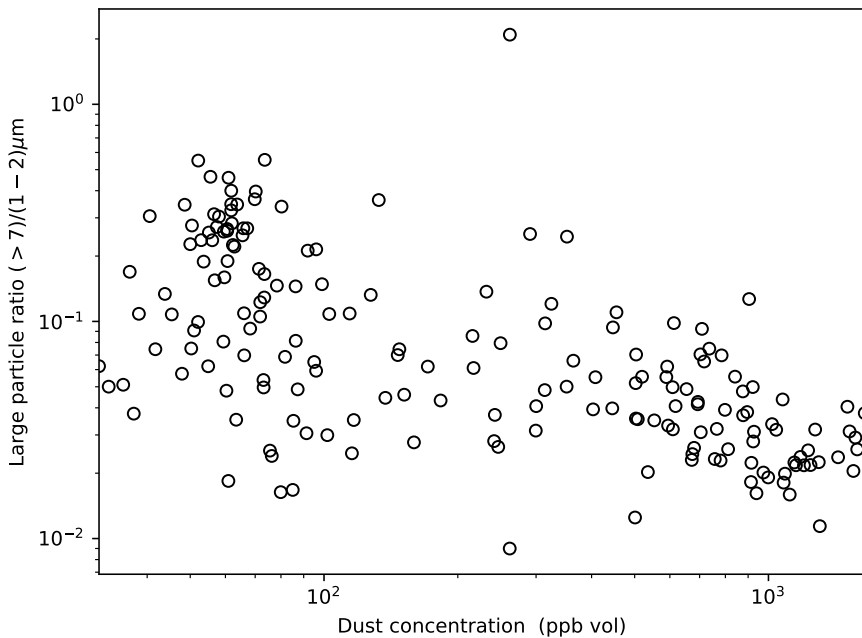

**Figure 11.** The ratio between small and large particle concentration by volume as a function of total dust concentration. The points are 10 cm sections from 533-554 m depth.

facing the light. This gives a unique $d_{\text{ext}}$ for each $d_{\text{vol}}$, and Equation 1 is replaced by

$$\frac{dP(d_{\text{ext}}|d_{\text{vol}})}{d\ln d_{\text{ext}}} = \delta(\ln d_{\text{ext}} - \ln(\alpha d_{\text{vol}})), \tag{G1}$$

where $\delta$ is the Dirac delta function and

$$\alpha = \sqrt{\frac{\pi}{4c}}. \tag{G2}$$

This value of $\alpha$ is calculated by letting $c$ be the aspect ratio of the rectangle, $d_{\text{ext}}$ the long side and $d_{\text{vol}}$ the diameter of a circle with the same radius as the rectangle. In effect, Equation G1 means that the distribution of $d_{\text{ext}}$ is equal to the distribution of $d_{\text{vol}}$, just with $d_{\text{ext}} = \alpha d_{\text{vol}}$. If a prolate particle is located at $x = 0$ and $y \neq 0$, all rotation angles relative to the light are equally likely. It can therefore be modelled like the oblates. We assume that all positions in the $x, y$ plane are equally likely. For $x \neq 0$ and $y \neq 0$, $\frac{dP(d_{\text{ext}}|d_{\text{vol}})}{d\ln d_{\text{ext}}}$ is a combination of Equation 1 and G1. The combination of the equations depend on $x$ and $y$ in a non-trivial manner. However, we assume that the aspect ratio derived from the combination will lie between the aspect ratios derived from Equation 1 and G1. To approximate the exact combination, we take the mean of the two distributions. Using Equation 3, modelled Abakus data can be generated from the Coulter Counter data, similar to what we did for the oblate particles. The best correspondence between modelled and real Abakus data is for $c = 0.34 \pm 0.04$ for the glacial and $c = 0.43 \pm 0.08$ for the Holocene. As the correct relative weight of Equation 1 and G1 in $\frac{dP(d_{\text{ext}}|d_{\text{vol}})}{d\ln d_{\text{ext}}}$ is uncertain, an upper bound on the aspect ratio

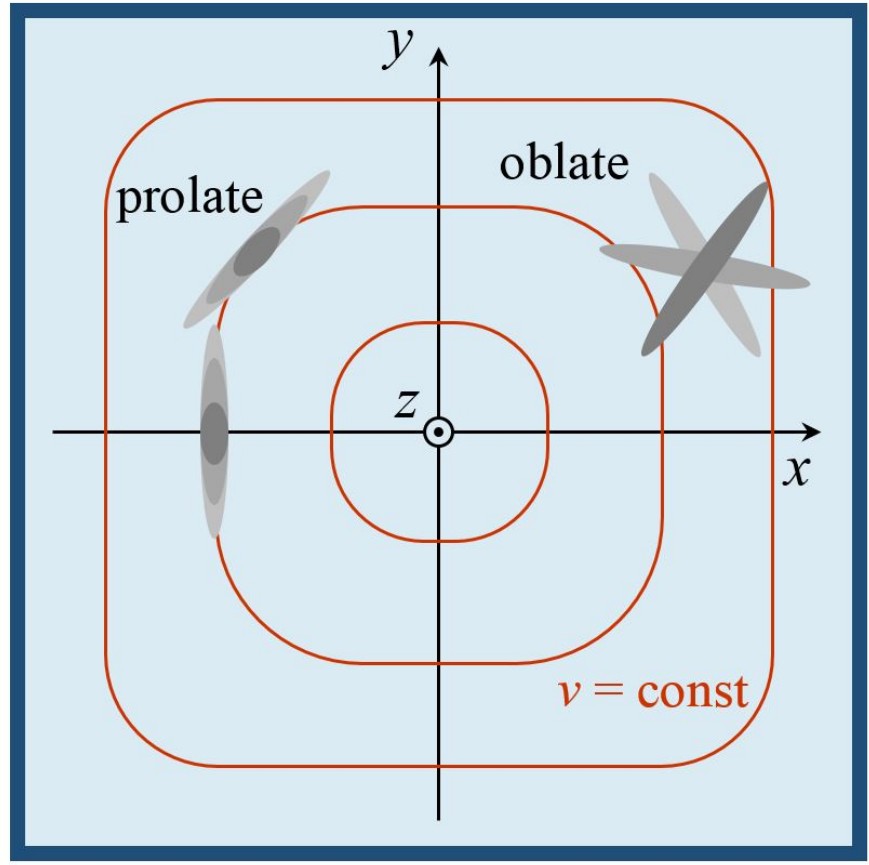

**Figure 12.** The Abakus flow cell. The flow is in the $z$ direction and the laser beam is in the $x$ direction. The red curves are contours of constant flow velocity. Oblate particles are free to rotate in the $x, y$ plane, while prolates can rotate in a plane of constant flow.

can be calculated by using only Equation G1. This gives $c = 0.37 \pm 0.03$ for the glacial and $c = 0.45 \pm 0.06$ for the Holocene. The uncertainty on $c$ arising from a wrong relative weight of Equation 1 and G1 is therefore smaller than the uncertainty from other sources, and can be neglected. In conclusion, the aspect ratio derived from comparing Abakus to Coulter Counter data is less extreme when prolate rather than oblate particles are assumed.

ADDA simulations of prolate particles can also be used to extract an aspect ratio from the SPES data. This is rather artificial, as the SPES data shows that the particles are oblate. However, we do it to show that adding a fraction of prolates does not change the fitted aspect ratio significantly. As described in section 3.3, the mean optical thickness $\rho$ is calculated as function of extinction cross section $\sigma_{\text{ext}}$ for different aspect ratios $c$ (Figure 13). These curves are interpolated to generate a contour plot of $c$ in $\rho, \sigma_{\text{ext}}$ space. The mean $\rho$ curve is then calculated for the aspect ratios 0.25, 0.33 and 0.50 for prolate particles. By fitting these mean $\rho$ curves to the oblate aspect ratio contour, it is found that they correspond to oblate particles of aspect ratio 0.23, 0.28 and 0.41 respectively. The fit is only performed for $\sigma_{\text{ext}}$ values between the 0.25 and 0.75 quantiles of the SPES data, ie.

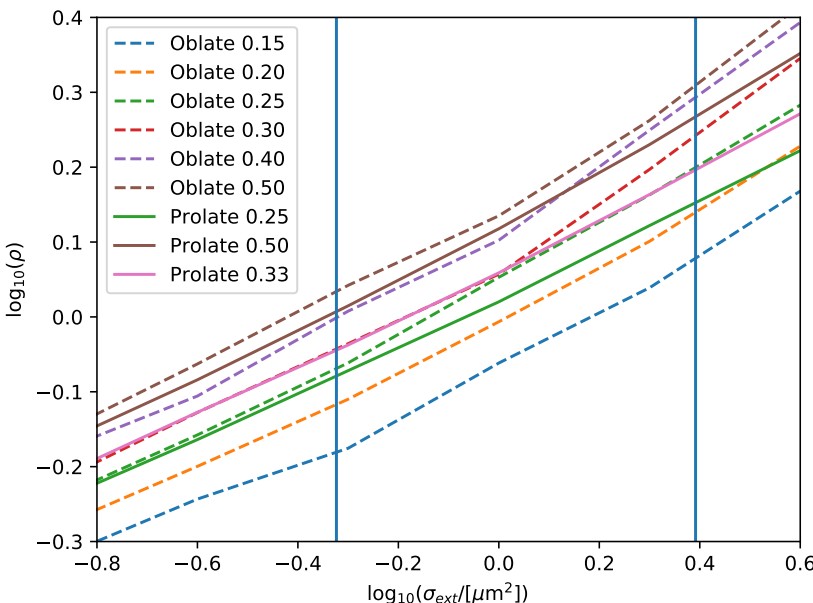

**Figure 13.** Mean optical thickness curves for oblate and prolate particles for different aspect ratios indicated in the legend.

between the blue lines of Figure 6. By interpolating between the prolate/oblate aspect ratio pairs, it is found that the prolate
5   aspect ratios corresponding to the oblate aspect ratios of 0.33 and 0.39 of the RECAP glacial and Holocene samples are 0.39
and 0.47 respectively (Figure 14). The ADDA simulations therefore predict less extreme aspect ratios for prolates than for
oblates, similarly to the Abakus/Coulter Counter comparison. This analysis is however artificial in the sense that the RECAP
samples are dominated by oblates and not prolates.

### Appendix H:  Abakus calibration scheme

10   The Abakus has to be calibrated in two steps: first to yield the extinction cross section (Section 3.2), and then to account for an
aspect ratio different from 1 (Section 3.5). Here is a step-by-step guide to the calibration.

1. Extinction calibration

    (a) Measure standard polystyrene spheres with the Abakus, preferably at least 5 different sizes. For polar ice, the range
        1-20 μm is appropriate.

    (b) The spheres have a true diameter given by the manufacturer and now also a measured diameter given by the Abakus.
5        The theoretical extinction diameter as a function of true diameter is shown in Figure 2. Divide the measured data
        by the theoretical curve to get a ratio for each data point.

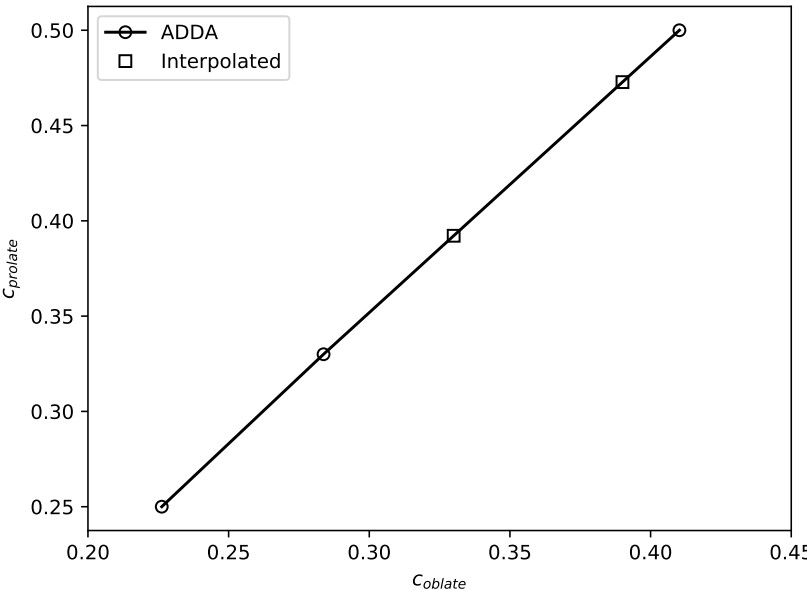

**Figure 14.** ADDA simulated prolate aspect ratios for the corresponding oblate aspect ratios, derived from Figure 13. The squares are at $c_{\text{oblate}}$ values of $0.33$ and $0.39$, interpolated between the simulated values.

    (c) Connect each ratio by interpolation or preferably by fitting a smooth function to the ratio versus the measured diameter. This gives a continuous ratio function.

    (d) Multiply the Abakus bin boundaries by the ratio function. The Abakus data with the new bin boundaries are now a histogram of extinction diameters.

2. Aspect ratio calibration

    (a) If the aspect ratio is known, for example from SPES, multiply the Abakus bin boundaries by the cubic root of the aspect ratio. The Abakus data is now calibrated.

    (b) If the aspect ratio is not known, use Coulter Counter data to calibrate the Abakus by the following optimisation algorithm. Multiply the Abakus bin boundaries by a variable $c$ to get nominally calibrated Abakus data.

    (c) Take the logarithm of both Coulter Counter and Abakus data, subtract the two and take the absolute value. The sum of the absolute values is the badness of the optimisation.

    (d) Vary $c$ to minimise badness. The nominally calibrated Abakus data corresponding to minimal badness may be used as calibrated Abakus data.

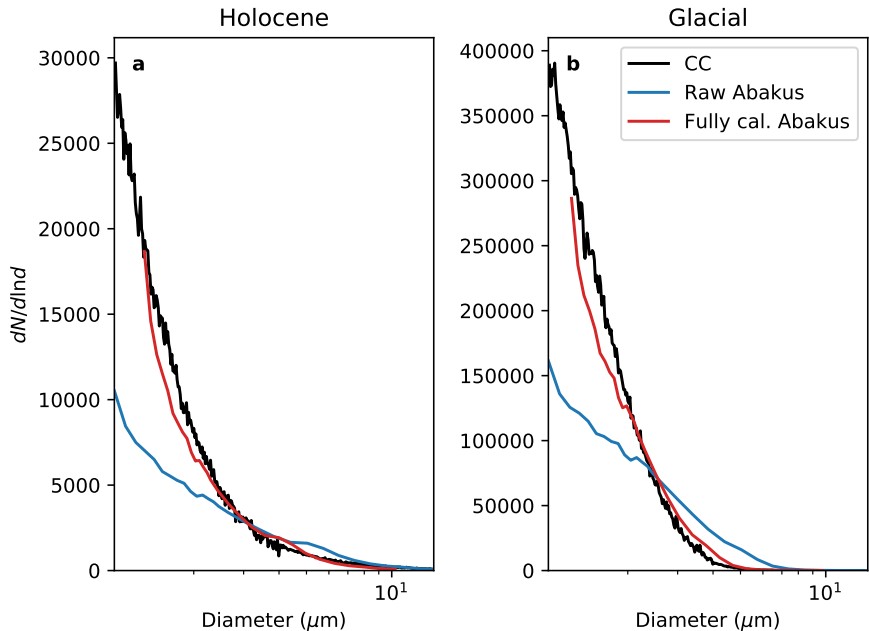

**Figure 15.** Figure 5a,b on a linear vertical scale.

## Appendix I: Detection limit

The number size distribution is a decreasing function even for the smallest detectable diameters (Figure 15). This means that there are most likely many more particles below the detection limit than above. The total counts therefore depends heavily on where the lower cutoff is. After calibration, the Coulter Counter and the calibrated Abakus give almost the same total number of particles for the same size range.

*Competing interests.* The authors declare that they have no conflict of interest.

*Acknowledgements.* The RECAP ice coring effort was financed by the Danish Research Council through a Sapere Aude grant, the NSF through the Division of Polar Programs, the Alfred Wegener Institute, and the European Research Council under the European Community's Seventh Framework Programme (FP7/2007-2013) / ERC grant agreement 610055 through the Ice2Ice project. The Centre for Ice and Climate is funded by the Danish National Research Foundation.

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
