# Peer review of "Particle shape accounts for instrumental discrepancy in ice core dust size distributions"

_Climate of the Past, 2017_

## Referee Comment (RC1) · Anonymous Referee #1 · 23 Dec 2017

This manuscript by Simonsen et al. is an important addition to the ice core dust community. The discrepancy between Abakus and Coulter Counter measurements of dust particle size have been discussed in the past but this study provides a clear description of the measurement differences due to the irregular shape of dust particles. More importantly, this study presents a calibration routine for adjusting the Abakus measurements thereby providing more accurate measurements and allowing for comparisons between different ice cores, and ice core lab groups. Some comments on the writing and clarity of the paper are below.

Comments: Page 1 Line 3: "depending on the type of sample." Can you be more specific here? do you mean mineral dust particles versus volcanic tephra?

Line 5: To accurately measure irregularly shaped dust particles, a calibration routine

based on standard spheres must be used.

Line 13: The dust has several properties that provide useful information of the past:

Line 15: The observed 100-fold decrease

Page 2 Line 3-4: offering faster measurement speed Line 5: and melted upon a gold coated melt head... Line 9: The Abakus instrument measures the intensity of laser light through a flow cell filled with the sample liquid. Line 10: The Abakus therefore measures the optical extinction cross section of the particle, and can measure particles in the range of 1-15 um. Line 12-13: Can you be more specific or explain more on the depth resolution of 3 mm? Line 14: What do you mean "thicker interval"? Line 17: The SPES instrument measures the extinction cross section (also measured by the Abakus) in addition to the optical thickness of the particles (Potenza et al., 2016). Line 25: Furthermore the ADDA simulations were used to show that the Mie scattering effects on the optical extinction cross-section for spherical particles do not affect ice core dust due to its irregular shape... Line 28: The measured samples are from the Renland Ice Cap Project (RECAP) ice core drilled during the summer of 2015... Line 31: The Holocene dust, similar to the old Renland core (Bory et al., 2003), is dominated by a local East Greenlandic source. Line 33: The volume mode of the glacial dust is 2 um versus 20 um for the Holocene dust, which is likely due to the increased transport size fractionation of glacial dust (Ruth et al., 2003).

Page 3 Line 4: We show here that the non-spherical shape of the particles... Line 15: higher melt rate Line 15-16: The Abakus was connected to the CFA system with a flow rate of 2 mL/min. Line 17: remove "during the measurement campaign." Line 18: ...was clogged by a large particle and required flushing... Line 19: The Abakus was initially calibrated to the diameters of polystyrene beads of 2, 5 and 10 um.

Page 4 Line 4: I'm not sure this sentence is necessary. Line 6: The Coulter Counter measured discrete samples 55 cm in length. Line 6-7: The measured ice consisted of an outer triangular piece 3 x 1 cm in cross section, which was broken lengthwise into

smaller pieces of 10 cm. Line 10: The samples were shaken prior to measurement in the 100 um Coulter Counter. Line 13: Selected samples representative of Holocene and glacial climates were measured by the Coulter Counter for this study. Line 30: please define all parameters in the equation

Page 5 Line 1: You should provide more information about what the extinction diameter actually is before the rest of the paragraph. You talk about it more in section 3.2 but it should really be explained here. Line 2: remove "Besides," Lines 6-8: The two consecutive sentences "For each volumetric diameter, there will therefore be a probability density function of extinction diameter. When the Abakus measures a particle, it gives a specific extinction diameter with a probability given by this probability density function." seem a bit redundant and this can be shortened. Line 8-9: The broadness of the probability density function for non-spherical particles results in a smoothing of the relationship between the extinction and volumetric diameters (Figure 3). Line 9 & 11: I think the word wiggles can be replaced with "scattering" or something like that. Line 12: The average extinction diameter is higher than the volumetric diameter due to the elongated particles measured (discussed further in section 3.4). Line 18: We have measured the diameters of five different. . .

Page 6: Line 1-2: This sentence also seems a bit redundant. . . Line 11-13: To compare the simulations and the data, the average of the logarithm of the simulated optical thickness as a function of extinction cross section was calculated. This average was then used as a least squares fit to the measured data, where the aspect ratio is the variable parameter. Line 13: is the refractive index of ice core dust "n"? That you talk about on page 7 line 2? If so, you should define it here.

Page 7 Line 4: Insert period after "parabolic fit" Line 6-7: The SPES is not sensitive in the full optical thickness range for low and high extinction cross sections, which would introduce a bias when fitting to the simulated data. Line 11: We define the geometric cross section diameter of a particle as. . .

Page 8 Figure 4 caption: delete one of the "log", should you use "logarithm" instead since you use it throughout the text elsewhere?

Page 10 Line 3-4: The discrepancy between the two instruments is systematic and exists because they measure two different properties of the particles: volume and extinction cross section.

Page 11 Line 7: . . .not the uncertainty of the mean." Line 9-10: I'm not sure you need to include that information here, I think its more important to talk about RECAP dust provenance. Line 12: Since large ice sheets are located far from typical dust sources, the dust extracted. . .

Page 12 Line 12: Moreover, by determining the aspect ratio, a more accurate size distribution can be obtained from the Abakus data.

I think that you can add a few lines in the conclusion section about what the implications of this research are, how can this model aid in more accurately determining ice core provenance?

---

## Referee Comment (RC2) · Anonymous Referee #2 · 27 Dec 2017

In this manuscript Simonsen and colleagues tackle the long-standing problem that the Klotz Abakus particle counting device yields different results from the established Coulter Counter method for ice-core dust. They argue that because of the asymmetric shape of natural dust particles, the Abakus sensor has to be calibrated using the extinction diameter and not the geometric diameter of particles. Since the CC measures the true particle volume but the Abakus a two-dimensional cross section, they combine the two measurements on ice core data from Greenland to infer the average aspect ratio of dust particles during Holocene and LGM sections of the record.

The method described in the manuscript is innovative and a logical continuation of the studies previously published by the author groups. I am not quite happy with the ice core data application in its current form, though. The authors seem to mix and match

parameters taken from various Antarctic and Greenland ice core dust publications. In addition, several assumptions are not well justified. This ultimately creates a result that may be very dependent on the specific parameters used. I therefore recommend major revisions before I can support the publication of this manuscript.

Major Comments:

In Chapter 3.1 you define a PDF that attributes a probability range of the extinction diameter as a function of the Volumetric diameter. How does this PDF come into play in the rest of the manuscript? Is it needed for the calibration? If not, it doesn't seem to be of use after that and maybe this chapter may not be necessary? In general, the method is a bit confused. Think of other groups that own an Abakus and want to calibrate their instrument using your method. Provide them with a clear set of instructions on how to do this.

I have the feeling that Chapter 3.3 is too short. There is very little text to explain a lot of material and as a consequence it is very difficult to understand. I think this section should be greatly expanded. But more concerning is the authors' claim that ice core dust refractive indexes vary between 1.52 and 1.55, citing Royer et al., 1983. These are not two limiting values, they are just two values found for Holocene and LGM ice. Moreover, they were calculated for Antarctic dust at 546 nm wavelength. This manuscript deals with Greenland dust and the Abakus laser has a wavelength of 680 nm. If the simulations are not too computationally intensive one could make a Monte-Carlo run with a whole range of values. Else, refractive indexes measured from RECAP particles should be used.

The authors claim in Page 10, line 2 that the Abakus counts 10 times more particles than the CC. That goes against the findings in Ruth et al., 2008: "Good correspondence (Rlog = 1.00 and clog = 0.92) is found also between the respective number concentrations" and against the findings in Fujii et al., 2003 and Lambert et al., 2012 who claim that coincidence loss will result in lower counts for the Abakus than the CC due

to several particles passing the laser beam at the same time. If the authors measured 10 times more particles with the Abakus than with the CC in the RECAP ice core, then they should explain why they get such opposite results from previous studies. I will assume that this is a tipo and the authors meant they measured 10 times less particles with the Abakus. This brings up another problem though. The much higher counting efficiency of the CC suggests that coincidence loss in the Abakus is the norm rather than the exception, and this will distort the size measurements in the Abakus. This aspect should be addressed in this paper as well.

Minor Comments:

Introduction: English is sub-standard. Please revise language.

Page 1, line 5: delete "leads" Page 1, line 6: What new calibration routine? Page 1, line 17: These references have nothing to do with climate models. Page 2, line 2: "...due to its sensitivity to electrical noise." That is the problem with coupling it to a CFA system? Please explain in more details. Page 2, line 3: CFA is not a technique to prepare samples. Page 2, line 32: Delete one occurrence of Bory et al., 2003 Page 3, lines 9-10: That is a big assumption. Either you show this is the case or you concentrate on the method. Figure 4: What's the green shading? Uncertainty? If so, how was it defined? Figure 5: I don't see how the calibration improves the Abakus data if the CC is the reference. The calibrated curve seems worse than the uncalibrated to me by eye. Maybe a plot of residuals and a SSE could provide a quantitative measure of improvement? Page 6, line 8: This reference does not support the assumption that the samples are dominated by oblates. There is one sentence about Antarctica, but I don't think results from Antarctica could be extended to Greenland, see my main comments. Also, how would the method perform if the sample was not dominated by oblates? Page 8, line 1: Again I don't think excluding prolates is justified unless you show the an analysis from the RECAP ice-core. Anyway, the method to calculate aspect ratios for both types of particles has been established by Potenza et al., 2016, so why exclude the prolates? Page 11, line 7-8: You only calculate aspect ratios of oblates in this study.

---

## Editor Comment (EC1) · EW Wolff (Editor) · 4 Jan 2018

You now have 2 review comments on your paper. Both are supportive of the idea behind the paper, but referee 2 in particular raises some important issues that need to be addressed. As you are aware the procedure is that you must respond to all the review comments (within 4 weeks of the discussion closing), and then I am asked to provide editorial guidance on whether you should submit a revised paper. However in this case, since the reviews are in early, it may be helpful if you have time to respond to the major comment about the difference in particle number between Coulter and laser methods early, as this would give time for a genuine interactive discussion between you and the reviewer, which might resolve the problem before you prepare a new version.

[Figure]

I would like to add my own comment on the issue raised by referee 2. From Figure 5 it is clear that your sttaement about Abakus giving 10 times more particles than Coulter is not a typo, and needs some clarification and explanation above what is in the paper now. It is, as the reviewer points out, much at odds with the finding for EDML (Antarctica) in Ruth et al (2008) who said (my annotations in [])"The [CC and LPD] data [for particle mass] have a very high correlation (Rlog ) 1.00); and the clog of 0.96 is very close to 1. Good correspondence (Rlog ) 1.00 and clog)0.92) is found also between the respective number concentrations (data not shown)."

If we compare in Fig 5 your Coulter and calibrated laser data, it seems as if the laser counts way more particles right across the size spectrum, so this is not just a question of it classing them in the wrong size range because of the particle shape. However then I am very confused by Figure 7. This seems to show a ratio (between calibrated Abakus and Coulter for the glacial) of a maximum 1.5 at about 5 microns, and below 1 at 2 microns and 8 microns. But in Figure 5 (which is apparently the same data), the ratio is clearly more than 10 at 5 microns, and more than 1 at all diameters. Please explain the apparent discrepancy between the two figures. Perhaps you have written calibrated and uncalibrated the wrong way round in the caption to Fig 5, although this cannot explain the glacial factor (>1 in Fig 5, «1 in Fig 7) at 10 microns? If the very large (factor 10) difference between the methods actually applies only to the uncalibrated Abakus, then at least this makes better sense, but leaves two issues for you to comment on:

(1) Assuming coincidence counting is not a huge issue in either method, the total number of particles should be the same in the uncalibrated Abakus, the calibrated Abakus and the Coulter. Is that the case? - the fact that one Abakus curve is ALWAYS above the Coulter curve makes me doubt that but it could just be a subtlety of how the integration works when one curve goes to lower diameter than another.

(2) You need to discuss why this problem (between uncalibrated Abakus and Coulter) apparently didn't show up at EDML (Ruth et al 2008). Does it only apply to Greenland?

I hope your comments on this might clarify at least the technical point about which curve is which and how to reconcile Figs 5 and 7, and thus allow the reviewer and me to understand the work better.

---

## Author Comment (AC1) · 2 Feb 2018

Dear Eric Wolff

Thanks for your interest in the manuscript and thorough and helpful comments. Below we have copied your text and responded to each section individually. For some of your comments, we have responded in the answer to Referee #2, and refer to the answers given there.

Attached is a document, where we respond point by point to your comments. At the end of document, there is revised version of the manuscript. Most of our replies to the referees refer to this revised manuscript.
Please also note the supplement to this comment:
https://www.clim-past-discuss.net/cp-2017-149/cp-2017-149-AC1-supplement.pdf

[Figure]

**Supplement:**

**EW Wolff (Editor)**

**ew428@cam.ac.uk**

**You now have 2 review comments on your paper. Both are supportive of the idea behind the paper, but referee 2 in particular raises some important issues that need to be addressed. As you are aware the procedure is that you must respond to all the review comments (within 4 weeks of the discussion closing), and then I am asked to provide editorial guidance on whether you should submit a revised paper. However in this case, since the reviews are in early, it may be helpful if you have time to respond to the major comment about the difference in particle number between Coulter and laser methods early, as this would give time for a genuine interactive discussion between you and the reviewer, which might resolve the problem before you prepare a new version.**

**I would like to add my own comment on the issue raised by referee 2. From Figure 5 it is clear that your sttaement about Abakus giving 10 times more particles than Coulter is not a typo, and needs some clarification and explanation above what is in the paper now. It is, as the reviewer points out, much at odds with the finding for EDML (Antarctica) in Ruth et al (2008) who said (my annotations in [])"The [CC and LPD] data [for particle mass] have a very high correlation (Rlog ) 1.00); and the clog of 0.96 is very close to 1. Good correspondence (Rlog ) 1.00 and clog)0.92) is found also between the respective number concentrations (data not shown)."**

We have replied to this in our reply to referee #2.

**If we compare in Fig 5 your Coulter and calibrated laser data, it seems as if the laser counts way more particles right across the size spectrum, so this is not just a question of it classing them in the wrong size range because of the particle shape. However then I am very confused by Figure 7. This seems to show a ratio (between calibrated Abakus and Coulter for the glacial) of a maximum 1.5 at about 5 microns, and below 1 at 2 microns and 8 microns. But in Figure 5 (which is apparently the same data), the ratio is clearly more than 10 at 5 microns, and more than 1 at all diameters. Please explain the apparent discrepancy between the two figures. Perhaps you have written calibrated and uncalibrated the wrong way round in the caption to Fig 5, although this cannot explain the glacial factor (>1 in Fig 5, «1 in Fig 7) at 10 microns? If the very large (factor 10) difference between the methods actually applies only to the uncalibrated Abakus, then at least this makes better sense, but leaves two issues for you to comment on:**

We apologise, both Figure 5 and 7 are quite misleading. We have merged them into a new figure (Figure 5), which is hopefully more clear.

**(1) Assuming coincidence counting is not a huge issue in either method, the total num- ber of particles should be the same in the uncalibrated Abakus, the calibrated Abakus and the Coulter. Is that the case? - the fact that one Abakus curve is ALWAYS above the Coulter curve makes me doubt that but it could just be a subtlety of how the inte- gration works when one curve goes to lower diameter than another.**

We have added an appendix on coincidence.

**(2) You need to discuss why this problem (between uncalibrated Abakus and Coulter) apparently didn't show up at EDML (Ruth et al 2008). Does it only apply to Greenland?**

We have replied to that in our reply to referee #2.

**I hope your comments on this might clarify at least the technical point about which curve is which and how to reconcile Figs 5 and 7, and thus allow the reviewer and me to understand the work better.**

Yes, we agree that it was unclear. We have tried to clarify in the new version.

[revised manuscript text omitted]
. To derive a volumetric size distribution from Abakus data, the Abakus data first have to be extinction calibrated (Section 3.2) so it gives extinction diameter instead of measured diameter (Section 3.2). Then an inverted Equation 3 has to be applied to account for the aspect ratio effect. The inversion cannot be done exactly with $\frac{dP(d_{\text{ext}}|d_{\text{vol}})}{d\ln d_{\text{ext}}}$ (Equation 1) in its current form. It therefore has to be done semi empirically. Since the modelled Abakus data fits the extinction
15  calibrated Abakus data (Figure 5), the discrepancy between the Abakus and Coulter Counter can be explained by wrong binning, and not by missing counts. For the aspect ratio inversion, it is therefore only necessary to shift the bins, and not change the counts in each bin. We propose to multiply the Abakus bins by $\sqrt[3]{c} = (0.73, 0.69)$ for the Holocene and glacial respectively, where $c$ is the aspect ratio. The result is the fully calibrated Abakus data seen in Figure 5. The multiplication factor is chosen to make the fully calibrated Abakus data consistent with the the Coulter Counter data. For the glacial volumetric distributions,
20  the mode of the Coulter Counter, the uncalibrated Abakus and the fully calibrated Abakus data are respectively 2.3, 5.0 and 2.7 µm.

**4   Discussion**

When the Abakus is calibrated using the true diameter of polystyrene spheres, it gives up to 10 times as many counts as the Coulter Counter for some particle sizes, when ice core samples are measured (Figure 5). This result is consistent with the
25  findings of Ruth et al. (2008, Figure 1a) who demonstrated that the data produced by the two instruments 
[revised manuscript text omitted]

---

## Author Comment (AC2) · 2 Feb 2018

**In this manuscript Simonsen and colleagues tackle the long-standing problem that the Klotz Abakus particle counting device yields different results from the established Coul- ter Counter method for ice-core dust. They argue that because of the asymmetric shape of natural dust particles, the Abakus sensor has to be calibrated using the ex- tinction diameter and not the geometric diameter of particles. Since the CC measures the true particle volume but the Abakus a two-dimensional cross section, they combine the two measurements on ice core data from Greenland to infer the average aspect ratio of dust particles during Holocene and LGM sections of the record.**

**The method described in the manuscript is innovative and a logical continuation of the studies previously published by the author groups. I am not quite happy with the ice core data application in its current form, though. The authors seem to mix and match parameters taken from various Antarctic and Greenland ice core dust publications. In addition, several assumptions are not well justified. This ultimately creates a result that may be very dependent on the specific parameters used. I therefore recommend major revisions before I can support the publication of this manuscript.**

**Major Comments:**

**In Chapter 3.1 you define a PDF that attributes a probability range of the extinction diameter as a function of the Volumetric diameter. How does this PDF come into play in the rest of the manuscript? Is it needed for the calibration? If not, it doesn't seem to be of use after that and maybe this chapter may not be necessary?**

The concept of a PDF representing the probability of an extinction diameter given by a volumetric diameter is described in more detail in Section 3.4. There it is used in the simplified 2D model, but it could in principle also be constructed for 3D particles, if their shape and orientation probability were known. A reference to Section 3.4 has been added to the text.

**In general, the method is a bit confused. Think of other groups that own an Abakus and want to calibrate their instrument using your method. Provide them with a clear set of instructions on how to do this.**

We have added a section 3.5, which gives a calibration instruction for the Abakus. We have also merged figure 5 and 7 into a new figure 5, which hopefully clarifies both figures. This includes a plot of a calibrated versus an uncalibrated Abakus volume size distribution compared to the corresponding Coulter Counter volume size distribution.

**I have the feeling that Chapter 3.3 is too short. There is very little text to explain a lot of material and as a consequence it is very difficult to understand. I think this section should be greatly expanded.**

We have now expanded it.

**But more concerning is the authors' claim that ice core dust refractive indexes vary between 1.52 and 1.55, citing Royer et al., 1983. These are not two limiting values, they are just two values found for Holocene and LGM ice. Moreover, they were calculated for Antarctic dust at 546 nm wavelength. This manuscript deals with Greenland dust and the Abakus laser has a wavelength of 680 nm. If the simulations are not too computationally intensive one could make a Monte-Carlo run with a whole range of values. Else, refractive indexes measured from RECAP particles should be used.**

In atmospheric dust studies such as Otto 2007, Highwood 2014 and Shettle 1979, they use a refractive index of dust at 670 nm of 1.53, but give no reference to where it is measured.

Sokolik, Andronova and Johnson 1993 measures 1.53-1.57 with a mean of 1.54 for atmospheric dust samples from Tadzhikistan, largely independent of wavelength within the visible range. They mention that Wahlstrom 1974 measures 1.55 at 633 nm. Grams 1974 measures a mean refractive index of 1.525 at 488 nm, but writes that there is some variance among the particles. Carlson and Benjamin 1980 find a refractive index of 1.54 for Saharan aerosols. Patterson and Gillette 1977 find a refractive index of around 1.547 at 670 nm of Saharan aerosols.

Some of these references have been added in the text. As they support using 1.52 and 1.55, we have not run new simulations.

**The authors claim in Page 10, line 2 that the Abakus counts 10 times more particles than the CC.**

The Abakus counts 10 times more in some size bins because the size bins are misaligned:

There are many more counts in the small than in the large bins. When the Abakus measures larger sizes than the Coulter Counter, all the small particle counts will be binned in a large particle bin, where there are only few Coulter Counter counts. Comparing the Abakus and Coulter Counter will show many more counts in the Abakus, simply because the bins are shifted.

**That goes against the findings in Ruth et al., 2008: "Good correspondence (Rlog = 1.00 and clog = 0.92) is found also between the respective number concen- trations"**

In their plot 1a, Ruth et al. show that CC and Abakus measure the same concentration. However, they have already shifted the Abakus bins empirically to make the size distributions fit the Coulter Counter distributions ("using CC data for the size calibration of the LPD"), as described in Ruth et al. 2003. This means that they cannot extract any information about absolute concentration from the Abakus, and they consequently do not do this. Instead, they comment on the "clog". The "clog" is the proportionality constant between the logarithm of the

Abakus and CC data. Any multiplicative factor between Abakus and CC does not affect "clog".

A constant scaling of the Abakus data would also not change the correlation, "Rlog". "good correspondence" therefore only means that the Abakus concentration is proportional to the Coulter Counter concentration, but not that they have the same value. They further write "the LPD is a reliable method to quantify variations of insoluble particle concentrations in ice cores", ie. it is good for variations, but they do not state whether it is good for absolute concentrations or not.

We have added the following line on this in the beginning of the Discussion section:

"This result is consistent with the findings of Ruth et al. (2008, Figure 1a) who demonstrated that the data produced by the two instruments are proportional over 25 four orders of magnitude, even if the absolute concentration results do not agree. "

**and against the findings in Fujii et al., 2003**

Fujii does not use an Abakus, but a different laser particle counter. His observed coincidence effects are therefore not directly comparable to the Abakus.

**and Lambert et al., 2012**

Lambert et al. write: "The regression between logarithmic CC concentration and logarithmic Bern LPD data is

$\log_{10}$ (CC mass concentration) = $(0.9084 \pm 0.0309) \cdot \log_{10}$ (Bern LPD data) $-$ $(1.3276 \pm 0.1076)$ with $r^2$ =0.85,n=519,

where CC mass concentration is given in ng $g^{-1}$ and the Bern LPD data in particles $ml^{-1}$. The regression between the logarithmic CC concentration and rescaled logarithmic CPH LPD data is

$\log_{10}$ (CC mass concentration) = $(1.633 \pm 0.089) \cdot \log_{10}$ (CPH LPD data) $+$ $(3.136 \pm 0.128)$ with $r^2$ =0.80,n=273"

They compare logarithms like Ruth et al.. Furthermore, they compare mass concentration with number concentration, so they cannot compare absolute concentration.

**who claim that coincidence loss will result in lower counts for the Abakus than the CC due to several particles passing the laser beam at the same time. If the authors measured 10 times more particles with the Abakus than with the CC in the RECAP ice core, then they should explain why they get such opposite results from previous studies. I will assume that this is a tipo and the authors meant they measured 10 times less particles with the Abakus. This brings up another problem though. The much higher counting efficiency of the CC suggests that coincidence loss in the Abakus is the norm rather than the exception, and this will distort the size measurements in the Abakus. This aspect should be addressed in this paper as well.**

We have added a supplementary section showing that coincidence effects are negligible, and referenced it in Section 2.1.

**Minor Comments:**

**Introduction: English is sub-standard. Please revise language.**

The introduction has been revised based on the comments from Referee #1.

**Page 1, line 5: delete "leads"**

OK.

**Page 1, line 6: What new calibration routine?**

Changed to:

"The irregular 5 shape means that a new calibration routine based on standard spheres is necessary for obtaining fully comparable data. This new calibration routine gives an increased accuracy on Abakus measurements, "

**Page 1, line 17: These references have nothing to do with climate models.**

No, We have now cited Mahowald et al. 1999 and Lambert et al. 2015.

**Page 2, line 2: "...due to its sensitivity to electrical noise." That is the problem with coupling it to a CFA system? Please explain in more details.**

Yes. It was found by Tobias Erhardt, but has not been described in a peer reviewed article. We have removed the sentence instead of expanding on the details.

**Page 2, line 3: CFA is not a technique to prepare samples.**

Changed to:

"CFA systems (Röthlisberger et al., 2000; Kaufmann et al., 2008) on the other hand are a common technique for analysing impurities in ice core samples, offering faster measurement speed and often higher resolution."

**Page 2, line 32: Delete one occurrence of Bory et al., 2003**

OK.

**Page 3, lines 9-10: That is a big assumption. Either you show this is the case or you concentrate on the method.**

Yes, the sentence is deleted.

**Figure 4: What's the green shading? Uncertainty? If so, how was it defined?**

The green shading is the uncertainty arising from the fit based on the uncertainty on the data points. It is used for calculating the uncertainty on the calibrated Abakus data in figure 5. We have added a line in the caption explaining this:

"The uncertainty on the fit (shading) is based on the uncertainty on the data points."

**Figure 5: I don't see how the calibration improves the Abakus data if the CC is the reference. The calibrated curve seems worse than the uncalibrated to me by eye. Maybe a plot of residuals and a SSE could provide a quantitative measure of improvement?**

No, it does not make the Abakus and CC data more alike. When the calibration is applied, the Abakus gives the extinction diameter. We see why this is confusing. We have merged figure 5 and 7 into a new figure 5, which hopefully clarifies this.

**Page 6, line 8: This reference does not support the assumption that the samples are dominated by oblates.**

No, it describes the measurement procedure. We have moved it to the previous sentence.

**There is one sentence about Antarctica, but I don't think results from Antarctica could be extended to Greenland, see my main comments.**

I assume you refer to page 11, line 5. We agree that one cannot assume that aspect ratios measured in Antarctica should be the same as in Greenland, as the source regions and transport processes are different. However, in our comparison, we merely state that the more extreme aspect ratio of Antarctic dust agrees with our hypothesis of aspect ratio fractionation during transport. We have removed the sentence about Antarctic dust provenance.

**Also, how would the method perform if the sample was not dominated by oblates? Page 8, line 1: Again I don't think excluding prolates is justified unless you show the an analysis from the RECAP ice-core. Anyway, the method to calculate aspect ratios for both types of particles has been established by Potenza et al., 2016, so why exclude the prolates? Page 11, line 7-8: You only calculate aspect ratios of oblates in this study.**

We have now explained why the samples are dominated by oblates. We have further added an appendix (G) discussing prolate particles, which show that the results do not differ significantly if prolate particles are hypothetically assumed:

"By comparing to SPES measurements of oblate and prolate particles in Villa et al. (2016) and Potenza et al. (2016), it was found that the samples are dominated by oblates. Prolates have a much narrower distribution of optical thickness than oblates, since their orientation is fixed by the flow. The absence of a superimposed prolate distribution indicates that no more than than 15% prolates are compatible with the measured SPES results. The following analysis therefore only focuses on oblates. For a similar analysis of prolates, see supplement G."

---

## Author Comment (AC3) · 2 Feb 2018

**This manuscript by Simonsen et al. is an important addition to the ice core dust com- munity. The discrepancy between Abakus and Coulter Counter measurements of dust particle size have been discussed in the past but this study provides a clear descrip- tion of the measurement differences due to the irregular shape of dust particles. More importantly, this study presents a calibration routine for adjusting the Abakus measure- ments thereby providing more accurate measurements and allowing for comparisons between different ice cores, and ice core lab groups. Some comments on the writing and clarity of the paper are below.**

**Comments: Page 1 Line 3: "depending on the type of sample." Can you be more specific here? do you mean mineral dust particles versus volcanic tephra?**

We basically mean that the deviation between the two depends on the aspect ratio of the particles. As we write that further down in the abstract, we have deleted it from this line.

**Line 5: To accurately measure irregularly shaped dust particles, a calibration routine based on standard spheres must be used.**

A calibration routine based on standard spheres is already used for the Abakus. However, another routine is necessary, since the particles are non-spherical. We

have written, based on reviewer 2's suggestion: "The irregular shape  means that a new calibration routine based on standard spheres is necessary."

**Line 13: The dust has several properties that provide useful information of the past: Line 15: The observed 100-fold decrease**

**Page 2 Line 3-4: offering faster measurement speed Line 5: and melted upon a gold coated melt head. . . Line 9: The Abakus instrument measures the intensity of laser light through a flow cell filled with the sample liquid. Line 10: The Abakus therefore measures the optical extinction cross section of the particle, and can measure particles in the range of 1-15 um.**

OK.

**Line 12-13: Can you be more specific or explain more on the depth resolution of 3 mm?**

In Bigler's paper, it means that  oscillating signals with a period below 3 mm cannot be discerned from a flat line, while the periodicity is visible for signals with a period above 3 mm.

**Line 14: What do you mean "thicker interval"?**

A thicker depth interval, for example 5 cm. It can therefore not resolve variations on shorter length scales. We have added "depth" in the sentence.

**Line 17: The SPES instrument measures the extinction cross section (also measured by the Abakus) in addition to the optical thickness of the particles (Potenza et al., 2016).**

We think it is the optical thickness that is the additional measurement, and not the extinction cross section. We have now changed it to:

"The SPES measures both the extinction cross section, which is also measured by the Abakus, and the optical thickness of the particles \citep{potenza2016}."

**Line 25: Furthermore the ADDA simulations were used to show that the Mie scattering effects on the optical extinction cross-section for spherical particles do not affect ice core dust due to its irregular shape. . . Line 28: The measured samples are from the Renland Ice Cap Project (RECAP) ice core drilled during the summer of 2015. . . Line 31: The Holocene dust, similar to**

the old Renland core (Bory et al., 2003), is dominated by a local East Greenlandic source. **Line 33: The volume mode of the glacial dust is 2 um versus 20 um for the Holocene dust, which is likely due to the increased transport size fractionation of glacial dust (Ruth et al., 2003).**

**Page 3 Line 4: We show here that the non-spherical shape of the particles... Line 15: higher melt rate Line 15-16: The Abakus was connected to the CFA system with a flow rate of 2 mL/min. Line 17: remove "during the measurement campaign." Line 18: . . .was clogged by a large particle and required flushing. . .**

OK.

**Line 19: The Abakus was initially calibrated to the diameters of polystyrene beads of 2, 5 and 10 um.**

In paragraph 3.2, we propose another calibration routine that also uses the same standards, but gives the extinction diameter and not the true diameter of the beads. We think it would be good to have this distinction already in the Methods section, and keep the word "correct".

**Page 4 Line 4: I'm not sure this sentence is necessary.**

No, we have removed it.

**Line 6: The Coulter Counter measured discrete samples 55 cm in length. Line 6-7: The measured ice consisted of an outer triangular piece 3 x 1 cm in cross section, which was broken lengthwise into smaller pieces of 10 cm. Line 10: The samples were shaken prior to measurement in the 100 um Coulter Counter. Line 13: Selected samples representative of Holocene and glacial climates were measured by the Coulter Counter for this study.**

OK.

**Line 30: please define all parameters in the equation**

We write that it is the relation between diameter and optical extinction cross section.

**Page 5 Line 1: You should provide more information about what the extinction diam- eter actually is before the rest of the paragraph. You talk about it more in section 3.2 but it should really be explained here.**

Yes, we have now written "Using the Amsterdam Discrete Dipole Approximation software (ADDA), we have calculated the extinction diameter for different particles. The extinction diameter is based on the optical extinction cross section. The optical extinction cross section is defined for a plane light wave interacting with a particle as the difference between the incoming and transmitted light intensity divided by the incoming light intensity and multiplied by the area of the plane incoming wave. For spherical particles much larger than …"

**Line 2: remove "Besides,"**

OK.

**Lines 6-8: The two consecutive sentences "For each volumetric diameter, there will therefore be a proba- bility density function of extinction diameter. When the Abakus measures a particle, it gives a specific extinction diameter with a probability given by this probability density function." seem a bit redundant and this can be shortened.**

OK, we have changed it to: "For each volumetric diameter, there will therefore be several possible Abakus measurements of the extinction diameter, described by a probability density function."

**Line 8-9: The broadness of the probability density function for non-spherical particles results in a smoothing of the relationship between the extinction and volumetric diameters (Figure 3).**

OK.

**Line 9 & 11: I think the word wiggles can be replaced with "scattering" or something like that.**

We have replaced it with "resonances".

**Line 12: The average extinction diameter is higher than the volumetric diameter due to the elongated particles measured (discussed further in section 3.4).**

We have replaced it with:

"The average extinction diameter is higher than the volumetric diameter since the measured dust particles are elongated (discussed further in section 3.4)."

**Line 18: We have measured the diameters of five different. . .**

OK.

**Page 6: Line 1-2: This sentence also seems a bit redundant. . .**

In a way, yes. But Line 1 relates to the previous line, while Line 2 relates to Lines 3 and 4. Maybe we are going into too much detail about the effect of the calibration? For now we have left it as it is, but we could shorten it down, leaving out the details.

**Line 11-13: To compare the simulations and the data, the average of the logarithm of the simulated optical thickness as a function of extinction cross section was calculated. This average was then used as a least squares fit to the measured data, where the aspect ratio is the variable parameter. Line 13: is the refractive index of ice core dust "n"? That you talk about on page 7 line 2? If so, you should define it here.**

**Page 7 Line 4: Insert period after "parabolic fit" Line 6-7: The SPES is not sensitive in the full optical thickness range for low and high extinction cross sections, which would introduce a bias when fitting to the simulated data. Line 11: We define the geometric cross section diameter of a particle as. . .**

**Page 8 Figure 4 caption: delete one of the "log", should you use "logarithm" instead since you use it throughout the text elsewhere?**

**Page 10 Line 3-4: The discrepancy between the two instruments is systematic and exists because they measure two different properties of the particles: volume and ex- tinction cross section.**

**Page 11 Line 7: . . .not the uncertainty of the mean." Line 9-10: I'm not sure you need to include that information here, I think its more important to talk about RECAP dust provenance. Line 12: Since large ice sheets are located far from typical dust sources, the dust extracted. . .**

**Page 12 Line 12: Moreover, by determining the aspect ratio, a more accurate size distribution can be obtained from the Abakus data.**

OK.

**I think that you can add a few lines in the conclusion section about what the implications of this research are, how can this model aid in more accurately determining ice core provenance?**

[revised manuscript text omitted]
}} = \int \frac{dP(d_{ext}|d_{vol})}{d\ln d_{ext}} \frac{dP(d_{vol})}{d\ln d_{vol}} d\ln d_{vol}. \tag{3}
$$

In section 3.2 we calibrated the Abakus such that $d_{meas} = d_{ext}$. Therefore $\frac{dP(d_{ext})}{d\ln d_{ext}}$ calculated here simulates Abakus measurements.

This can be used to find the aspect ratio of the particles just from the Coulter Counter-Abakus discrepancy. To do this, the sum of the square of the logarithm of the ratio between $\frac{dP(d_{\text{ext}})}{d\ln d_{\text{ext}}}$ calculated from the Coulter Counter data and the Abakus data was minimized with respect to the aspect ratio. This gives the aspect ratio where $\frac{dP(d_{\text{ext}})}{d\ln d_{\text{ext}}}$ is most consistent with the Abakus data, which is the most likely aspect ratio given the data. For the Holocene data this gave $c = 0.41 \pm 0.09$, while for the

5  glacial $c = 0.31 \pm 0.04$. The errors are propagated from the total errors on the calibrated Abakus data. There is however a large correlation between the errors on the Holocene and glacial data, so the error on the difference is only around 0.02, confirming a significant difference in aspect ratio between glacial and Holocene dust particles.

**3.5  Calibration of Abakus**

Equation 3 gives the extinction diameter size distribution (Abakus) given a measured volumetric size distribution (Coulter
10  Counter) and a known aspect ratio. However, often a measurement of the volumetric size distribution is desired, while only Abakus measurements are available. To derive a volumetric size distribution from Abakus data, the Abakus data first have to be extinction calibrated (Section 3.2) so it gives extinction diameter instead of measured diameter (Section 3.2). Then an inverted Equation 3 has to be applied to account for the aspect ratio effect. The inversion cannot be done exactly with $\frac{dP(d_{\text{ext}}|d_{\text{vol}})}{d\ln d_{\text{ext}}}$ (Equation 1) in its current form. It therefore has to be done semi empirically. Since the modelled Abakus data fits the extinction
15  calibrated Abakus data (Figure 5), the discrepancy between the Abakus and Coulter Counter can be explained by wrong binning, and not by missing counts. For the aspect ratio inversion, it is therefore only necessary to shift the bins, and not change the counts in each bin. We propose to multiply the Abakus bins by $\sqrt[3]{c} = (0.73, 0.69)$ for the Holocene and glacial respectively, where $c$ is the aspect ratio. The result is the fully calibrated Abakus data seen in Figure 5. The multiplication factor is chosen to make the fully calibrated Abakus data consistent with the the Coulter Counter data. For the glacial volumetric distributions,
20  the mode of the Coulter Counter, the uncalibrated Abakus and the fully calibrated Abakus data are respectively 2.3, 5.0 and 2.7 μm.

**4  Discussion**

When the Abakus is calibrated using the true diameter of polystyrene spheres, it gives up to 10 times as many counts as the Coulter Counter for some particle sizes, when ice core samples are measured (Figure 5). This result is consistent with the
25  findings of Ruth et al. (2008, Figure 1a) who demonstrated that the data produced by the two instruments 
[revised manuscript text omitted]

---

## Author Response (AR2)

**Suggestions for revision or reasons for rejection (will be published if the paper is accepted for final publication)**

This manuscript by Simonsen and colleagues has much improved. I accept most answers and rebuttals to my first comments. I am not yet convinced about a few aspects that should still be addressed, though and recommend minor revisions at this stage.

**Minor Comments:**

1. **At the end of section 3.1 you state that the Abakus, after calibration does not measure particles smaller than 1.8 um. This may be misunderstood. As I understand it, the Abakus can indeed measure particles below 1.8 um, but your calibration is not capable of going that small, possibly due to your choice of the smallest polystyrene sphere standard diameter of 1.5 um. If you had smaller standards could the calibration resolve smaller particles?**

The lowest diameter given by the uncalibrated Abakus is 1 um. In Figure 2 it is seen that the diameter ascribed by the Abakus of a polystyrene sphere of 1.5 um is only 1.1 um. The smallest measurable polystyrene sphere is therefore around 1.45 um. From the solid line in Figure 2 it is seen that the extinction diameter of a 1.45 um particle is 1.8 um.  The minimal measurable extinction diameter is therefore 1.8 um.

Dust particles have variable shape, which means that the Mie oscillations average out (Figure 3). For particles of variable morphology (variable shape with constant aspect ratio) but an aspect ratio of 1, the extinction diameter is equal to the volumetric diameter. For a different aspect ratio, the extinction diameter is larger than the volumetric diameter. It is therefore possible to measure smaller volumetric diameters for elongated particles. We have now specified that it is extinction diameter in the manuscript (Page 5, Line 18).

**2. In its present state, section 3.5 is just some copy-pasted text from other sections and absolutely useless for anyone wanting to apply your calibration to their data. These instructions don't need to be in the main text either. I suggest to put step-by-step instructions in the supplementary with both Abakus and CC data, and only Abakus data. You can use your data as an example.**

We have shortened the section and added step-by-step instructions to Supplement H.

**3. Let's go back to that factor 10 in the counting efficiency. I understand that if the calibration moves some counts to other bins, there may be a shift in the Abakus/CC ratio in certain bins. But not in the total counts. Your new figure 5d shows that in the raw data, the Abakus counts less small particles than the CC. This makes sense to me, as coincidence loss should be more probable for small particles than for large ones. Still, considering what we know about CC and Abakus detection limits, as well as coincidence loss, in my opinion the total Abakus count (summed over all bins) should to be lower than the total CC count. Could you add something about these two values in the discussion?**

The number size distribution is a decreasing function even for the smallest detectable diameters (See figure 15). This means that there are most likely many more particles below the detection limit than above. The total counts therefore depends heavily on where the lower cutoff is. For the same size range, the Coulter Counter and the calibrated Abakus do give the same total number of particles. We have changed the first paragraph of the Discussion and added Supplement I to describe this.

**Also, in Appendix F you state "If two small particles coincide in the Abakus detector, their combined shadow will make them look like a larger particle". You then compare small and large particle concentrations to total concentrations to conclude that coincidence loss is negligible. Your assumption is incorrect. The problem is not that two small particles combine to form one large. Loss occurs when one particle crosses the beam, and any smaller particle crossing at the same time (on either side of the large particle) will not be counted. Your method to show that coincidence loss in negligible is therefore incorrect or at least insufficient.**

The flow cell is 250 um wide. The laser light covers the whole cross section of the cell and is 1.5 um thick. It is therefore more like a sheet than a beam (see the figure at the end of this document, which is from Ruth et al., Ann. Glac. 2002). This means that the probability that two particles cross the laser sheet at the same time next to each other is much higher than behind each other, roughly 1-$d$/250um, where $d$ is the particle diameter.
The diameter assigned to a particle is based on the maximum peak height of the attenuation. Two particles passing next to each other give the sum of their individual shadows, and therefore a larger peak height than each of them individually. They are therefore detected as a larger particle.

If particles hiding behind each other was a significant problem, then particles passing next to each other would happen 20-100 times more often. Our appendix F shows that particles passing next to each other is a negligible phenomenon, so particles hiding behind each other is therefore negligible.

**4. Page 6, line 1: I think you mean Figure 4, not 5.**

No, specifically we mean the orange line in subfigures a and b. We have added that to the manuscript.

[Figure]

Fig. 2. Detection cell of the laser sensor (schematic). The cross section of the cell is 230 μm × 250 μm; the laser beam is 250 μm × 1.5 μm wide.